# Rethinking Calibration of Deep Neural Networks: Do Not Be Afraid of Overconfidence

**Deng-Bao Wang,**[1,2] **Lei Feng,**[3] **Min-Ling Zhang**[1,2*]

[1]School of Computer Science and Engineering, Southeast University, Nanjing 210096, China
[2]Key Laboratory of Computer Network and Information Integration (Southeast University),
Ministry of Education, China
[3]College of Computer Science, Chongqing University, Chongqing, 400044, China
wangdb@seu.edu.cn, lfeng@cqu.edu.cn, zhangml@seu.edu.cn

## Abstract

Capturing accurate uncertainty quantification of the predictions from deep neural networks is important in many real-world decision-making applications. A reliable predictor is expected to be accurate when it is confident about its predictions and indicate high uncertainty when it is likely to be inaccurate. However, modern neural networks have been found to be poorly calibrated, primarily in the direction of overconfidence. In recent years, there is a surge of research on model calibration by leveraging implicit or explicit regularization techniques during training, which achieve well calibration performance by avoiding overconfident outputs. In our study, we empirically found that despite the predictions obtained from these regularized models are better *calibrated*, they suffer from not being as *calibratable*, namely, it is harder to further calibrate these predictions with post-hoc calibration methods like temperature scaling and histogram binning. We conduct a series of empirical studies showing that overconfidence may not hurt final calibration performance if post-hoc calibration is allowed, rather, the penalty of confident outputs will compress the room of potential improvement in post-hoc calibration phase. Our experimental findings point out a new direction to improve calibration of DNNs by considering main training and post-hoc calibration as a unified framework.

## 1 Introduction

Modern over-parameterized deep neural networks (DNNs) have been shown to be very powerful modeling tools for many prediction tasks involving complex input patterns [37]. In addition to obtaining accurate predictions, it is also important to capture accurate quantification of prediction uncertainty from deep neural networks in many real-world decision-making applications. A reliable predictive model should be accurate when it is confident about its predictions and indicate high uncertainty when it is likely to be inaccurate. However, modern DNNs trained with cross-entropy (CE) loss, despite being highly accurate, have been recently found to predict poorly calibrated probabilities, unlike traditional models trained with the same objective [4]. The overconfident predictions of DNNs could cause undesired consequences in safety-critical applications such as medical diagnosis and autonomous driving. Bayesian DNNs, which indirectly infer prediction uncertainty through weight uncertainties, have innate abilities to represent the model uncertainty [2, 16]. But training and inferring those bayesian models are computationally more expensive and conceptually more complicated than non-bayesian models, and their performance depends on the form of approximation made due to computational constraints. Therefore, the study on uncertainty calibration of *deterministic* DNNs is important for both development practice and the perspective of understanding DNNs.

Post-hoc calibration addresses the miscalibration problem by equipping a given neural network with an additional parameterized calibration component, which can be tuned with a hold-out validation

---

*Corresponding author

35th Conference on Neural Information Processing Systems (NeurIPS 2021).

dataset. Guo *et al.* [4] experimented with several classical calibration fixes and found that simple post-hoc methods like Temperature Scaling (TS) [25] and Histogram Binning (HB) [33] are significantly effective for DNNs. The authors of [10] and [27] proposed to learn linear and non-linear transformation functions to rescale the original output logits respectively. Gupta *et al.* [5] proposed to obtain a calibration function by approximating the empirical cumulative distribution of output probabilities via splines. Kumar *et al.* [11] proposed to integrate TS with HB to achieve more stable calibration performance. Patel *et al.* [24] proposed a mutual information maximization-based binning strategy to solve the severe sample-inefficiency issue in HB.

Recently, there is another line of research which presents a possibility of improving the calibration quality of deterministic DNNs via regularization during training. Guo *et al.* [4] found that training DNNs with strong *weight decay*, which used to be the predominant regularization mechanism for training neural networks, has a positive impact on calibration. Müller *et al.* [19] showed that training models using the standard CE loss with label smoothing [28], instead of one-hot labels, has a very favourable effect on model calibration. Mukhoti *et al.* [18] proposed to improve uncertainty calibration by replacing the conventionally used CE loss with the focal loss proposed in [14] when training DNNs. It is important to note that CE loss with label smoothing and focal loss can be considered as standard CE with an additional maximum-entropy regularizer, which means minimizing these losses is equivalent to minimizing CE loss and maximizing the entropy of the predicted distribution simultaneously [18, 17]. Following these studies, a recent work [7] explored several explicit regularization techniques for improving the predictive uncertainty calibration directly.

In this paper, we conduct an empirical study showing that despite the predictions obtained from the regularized models are well *calibrated*, they suffer from worse *calibratable*, namely, it is harder to further improve the calibrate performance with post-hoc calibration methods like temperature scaling and histogram binning. We found that the regularization works by simply aligning the average confidence of the whole dataset to the accuracy with some specific regularization strengths, and cannot achieve fine-grained calibration. The comparison results show that when post-hoc calibration methods are allowed, the standard CE loss yields better calibration performance than those regularization methods. The extended experiments demonstrate that regularization will make DNNs lose the important information about the hardness of samples, which results in compressing the room of potential improvement by post-hoc calibration. Based on the experimental findings, we raise a natural question: can we design new loss functions in the opposite direction of these regularization methods to further improve the calibration performance? To this end, we propose *inverse focal loss*, and empirically found that it can learn more calibratable models in some cases compared with the CE loss, though it causes severer overconfidence problem without post-hoc calibration. Most importantly, our findings show that overconfidence of DNNs is not the nightmare in uncertainty qualification and point out a new direction to improve the calibration of DNNs by considering main training and post-hoc calibration as a unified framework.

## 2 Preliminaries

Let $\mathcal{Y} = \{1, ..., K\}$ denote the label space and $\mathcal{X} = \mathbb{R}^d$ denote the feature space. Given a sample $(\boldsymbol{x}, y) \in \mathcal{X} \times \mathcal{Y}$ sampled from an unknown distribution, a learned neural network classifier $f^\theta : \mathcal{X} \to \Delta^K$ can produce a probability distribution for $\boldsymbol{x}$ on $K$ classes, where $\Delta^K$ denotes the $K-1$ dimensional unit simplex. Here we assume $\boldsymbol{f}^\theta$ as a composition of a non-probabilistic $K$-way classifier $\boldsymbol{g}^\theta$ and a softmax function $\sigma$, i.e. $\boldsymbol{f}^\theta = \boldsymbol{g}^\theta \circ \sigma$. For a query instance $\boldsymbol{x}$, $\boldsymbol{f}^\theta$ gives its probability of assigning it to label $i$ as $\frac{\exp(g_i^\theta(\boldsymbol{x}))}{\sum_{k=1}^K \exp(g_k^\theta(\boldsymbol{x}))}$, where $g_i^\theta(\boldsymbol{x})$ denotes the $i$-th element of the logit vector produced by $\boldsymbol{g}^\theta$. Then, $\hat{y} := \arg\max_i f_i^\theta(\boldsymbol{x})$ can be returned as the predicted label and $\hat{p} := \max_i f_i^\theta(\boldsymbol{x})$ can be treated as the associated confidence score.

**Expected Calibration Error (ECE)** For a well-calibrated model, $\hat{p}$ is expected to represent the true probability of correctness. Formally, a *perfectly* calibrated model satisfies $\mathbb{P}(\hat{y} = y | \hat{p} = p) = p$ for any $p \in [0, 1]$. In practice, ECE [20] is a commonly used calibration metric from finite samples. It works by firstly grouping all samples (let $n$ denote the number of samples) into $M$ equally interval bins $\{B_m\}_{m=1}^M$ with respect to their confidence scores, then calculating the expected difference between the accuracy and average confidence: $\mathrm{ECE} = \sum_{m=1}^M \frac{|B_m|}{n} |\mathrm{acc}(B_m) - \mathrm{avgConf}(B_m)|$.

**Temperature Scaling** By scaling the logits produced by $\boldsymbol{g}^\theta$ with a temperature $T$, the sharpness

of output probabilities can be changed. Formally, after adding TS, the new prediction confidence can be expressed as: $\hat{p} = \max_i \frac{\exp(g_i^\theta(\boldsymbol{x})/T)}{\sum_{k=1}^K \exp(g_k^\theta(\boldsymbol{x})/T)}$. The temperature softens the output probability with $T > 1$ and sharpens the probability with $T < 1$. As $T \rightarrow 0$, the output probability collapses to one-hot vector. As $T \rightarrow \infty$, the output probability approaches to a uniform distribution. After training of the model, $T$ can be tuned on a hold-out validation set by optimization methods.

**Histogram Binning** is a non-parametric calibration approach. Given an uncalibrated model, all the prediction confidences of validation samples can be divided into mutually exclusive $N$ bins $\{B_n\}_{n=1}^N$ according to a set of intervals $\{I_n\}_{n=1}^{N+1}$ which partitions $[0, 1]$. Each bin is assigned a confidence score $\eta$, which can be simply set to the corresponding accuracy of samples in each bin. If the uncalibrated confidence $\hat{p}$ of a query instance falls into bin $B_n$, then the calibrated confidence is $\eta_n$. The bins can be chosen by two simple schemes: *equal size binning* (uniformly partitioning the probability interval in $[0, 1]$) and *equal mass binning* (uniformly distributing samples over bins). Note that although the HB scheme is simple to implement and was demonstrated to achieve good calibration results in some datasets, it makes the predictor only produce very sparse confidence distribution, and compromises the many legitimately confident predictions.

## 3 Regularization in Neural Networks for Calibration

In recent years, there is a surge of research on model calibration by leveraging implicit or explicit regularization techniques during training of DNNs, which makes better calibrated predictions by avoiding the overconfident outputs. In this section, we firstly review three representative regularization methods and then empirically show their improvements on ECE compared with the baseline.

**Label Smoothing** is widely used as a means to reduce overfitting of DNNs. The mechanism of LS is simple: when training with CE loss, the one-hot label vector $\boldsymbol{y}$ is replaced with *soft* label vector $\widetilde{\boldsymbol{y}}$, whose elements can be formally denoted as $\widetilde{y}_i = (1 - \epsilon)y_i + \epsilon/K, \forall i \in \{1, ..., K\}$, where $\epsilon > 0$ is a strength coefficient. Müller *et al.* [19] demonstrated that label smoothing implicitly calibrates DNNs by preventing the networks from becoming overconfident. Let $\mathcal{L}_{ce}$ denote the CE loss, then the following equation holds:

$$\mathcal{L}_{ce}(\widetilde{\boldsymbol{y}}, \boldsymbol{f}^\theta) = (1 - \epsilon)\mathcal{L}_{ce}(\boldsymbol{y}, \boldsymbol{f}^\theta) + \epsilon\mathcal{L}_{ce}(\boldsymbol{u}, \boldsymbol{f}^\theta) \tag{1}$$

This can be simply proved. Therefore, minimizing CE loss between smoothed labels and the model outputs is equivalent to adding a confidence penalty term, i.e., a weighted CE loss between the uniform distribution $\boldsymbol{u}$ and the model outputs, to the original CE loss.

$L_p$ **Norm in the Function Space** is one of the explicit regularization methods for calibration investigated by the recent work [7]. For a real number $p \geq 1$, the $L_p$ Norm of a vector $\boldsymbol{z}$ with dimension $n$ can be expressed as: $\|\boldsymbol{z}\|_p = (\sum_{i=1}^n |z_i|^p)^{1/p}$. By adding $L_p$ Norm of logits $\boldsymbol{g}^\theta$ with a weighting coefficient $\alpha$ into final objective function, i.e. $\mathcal{L}_{L_p}(\boldsymbol{y}, \boldsymbol{f}^\theta) = \mathcal{L}_{ce}(\boldsymbol{y}, \boldsymbol{f}^\theta) + \alpha \|\boldsymbol{g}^\theta\|_p$, the function complexity of neural networks can be directly penalized during training.

**Focal Loss** is originally proposed to address the class imbalance problem in object detection. By reshaping the standard CE loss through weighting loss components of all samples according to how well the model fits them, focal loss focuses on fitting hard samples and prevents the easy samples from overwhelming the training procedure. Formally, for classification tasks where the target distribution is one-hot encoding, it is defined as: $\mathcal{L}_f = -(1 - f_y^\theta)^\gamma \log f_y^\theta$, where $\gamma$ is a predefined coefficient. Mukhoti *et al.* [18] found that the models learned by focal loss produce output probabilities which are already very well calibrated. Interestingly, they also showed that focal loss is an upper bound of the regularized KL-divergence, which can be expressed formally as follows:

$$\mathcal{L}_f \geq \text{KL}(\boldsymbol{y}||\boldsymbol{f}^\theta) - \gamma\text{H}(\boldsymbol{f}^\theta) \tag{2}$$

where $\text{H}(\boldsymbol{p})$ denotes the entropy of distribution $\boldsymbol{p}$. This upper bound property shows that replacing the CE loss with focal loss has the effect of adding a maximum-entropy regularizer.

### 3.1 Empirical Comparison

We conduct a comparison study of the above regularization methods on four commonly used datasets. We train ResNet-32 [6] models on SVHN [21], CIFAR-10/100 [9] and train a 8-layer 1D-CNN

Table 1: Comparison results (mean±std) of ECE (%) with $M = 15$ and predictive accuracy (%) over 5 random runs. The values with underline in first row represent the chosen coefficients of each regularization method on four datasets according to the ECE on test data.

| | | Cross-Entropy | Label Smoothing 0.01/0.05/0.09/0.09 | $L_1$ Norm 0.01/0.05/0.01/0.01 | Focal Loss 1/3/5/5 |
|---|---|---|---|---|---|
| SVHN | ECE | 3.03±0.16 | 1.84±0.19 | 1.85±0.04 | 1.01±0.21 |
| | Accuracy | 95.00±0.27 | 95.21±0.23 | 95.29±0.13 | 94.77±0.19 |
| CIFAR-10 | ECE | 6.43±0.22 | 2.72±0.32 | 2.93±0.39 | 3.00±0.26 |
| | Accuracy | 90.46±0.23 | 90.09±0.41 | 90.06±0.59 | 87.84±0.17 |
| CIFAR-100 | ECE | 19.53±0.36 | 2.27±0.48 | 8.07±0.44 | 2.34±0.35 |
| | Accuracy | 64.64±0.43 | 63.73±0.67 | 63.07±0.29 | 60.36±0.44 |
| 20 Newsgroups | ECE | 20.82±0.93 | 5.85±0.64 | 13.31±0.56 | 3.82±0.51 |
| | Accuracy | 72.85±0.89 | 72.81±0.26 | 73.61±0.80 | 59.17±1.81 |

model on 20Newsgroups [13], using the standard CE loss and the above regularized losses respectively, with state-of-the-art learning policy settings (see implementation details in Appendix). For Norm regularization, we use $L_1$ Norm, which has been shown effective for calibration despite its simple form [7]. Table 1 shows the comparison of these methods. Note that for each of the above three regularization methods, there is a coefficient, i.e., $\epsilon$, $\alpha$ and $\gamma$, controlling the strength of regularization. We conduct experiments using these methods with the following coefficient settings: $\{0.01, 0.03, 0.05, 0.07, 0.09\}$ for label smoothing, $\{0.001, 0.005, 0.01, 0.05, 0.1\}$ for $L_1$ Norm and $\{1, 3, 5, 7, 9\}$ for focal loss. And we choose the best coefficient for each method and dataset, according to their ECE directly on test data.

From the results of Table 1, it is obvious that the regularization methods significantly decrease the ECE on all datasets, compared with the standard CE loss. The prediction accuracy results are also reported. When the strength coefficients of $L_1$ Norm and focal loss are large, their predictive performances are harmed. Especially, $L_1$ Norm fails on CIFAR-100 and 20Newsgroups when $\alpha \geq 0.05$, thus $\alpha$ is chosen from $\{0.001, 0.005, 0.01\}$ on these two datasets.

## 4 Does Regularization Really Help Calibration?

As shown by the above empirical results, the regularization methods *do* help the calibration of DNNs during training, especially alleviate the overconfidence issue caused by the standard CE loss. In this section, we empirically investigate their calibration performance when integrating them with post-hoc calibration. After training, we use the post-hoc methods TS and HB to further calibrate the output probabilities. For TS, we simply search the best temperature in the temperature pool $\{0.01, 0.02..., 10\}$ on the validation set (see data splits in Appendix). For HB, we use equal size binning scheme on the top-1 prediction of all classes with bin number set as 15. The experimental details used in this section are the same with those in Section 3.

**Comparison Results** Table 2 shows the comparison results of ECE with the help of TS and HB. We can see that: (1) The standard CE loss achieves the best calibration performance on most cases. (2) The searched temperatures of models trained with the CE loss are significantly higher than those of other losses, which indicates that CE loss causes higher predictive confidences. These results demonstrate that despite the regularized models can produce better calibrated predictions, it is harder to further improve them with post-hoc calibration methods after main training. In other words, the penalty of confident predictions will compress the room of potential improvement by post-hoc methods. We also conduct experiments on CIFAR-10 and CIFAR-100 using a deeper model ResNet-110 and similar comparison results are obtained (see Appendix Table A).

**Coefficient Sensitivity** Figure 1(a), 1(b) and 1(c) illustrate the ECEs of these three regularization methods with varied coefficient strengths. As we can see, since the complexity of the used datasets are different, the best coefficients of these methods markedly vary across the datasets, and a small change on these coefficients may cause large ECE increase. This means that we need to carefully choose the coefficient of each method when employing them on new datasets, to achieve good calibration. We can also observe that for SVHN, on which the accuracy is highest among four datasets, the regularization methods obtain lowest ECE with small coefficients. For CIFAR-100 and 20Newsgroups, on which the

Table 2: Comparison results (mean±std) of ECE (%) with $M = 15$ over 5 random runs. The coefficients of the regularization methods on each dataset are same with those in Table 1. ▲/▲ and ▼/▼ indicate that the average ECE of regularization methods are higher and lower than standard CE, where ▲ and ▼ are based on two-sample t-test at 0.05 significance level.

| | | Cross-Entropy | Label Smoothing 0.01/0.05/0.09/0.09 | $L_1$ Norm 0.01/0.05/0.01/0.01 | Focal Loss 1/3/5/5 |
|---|---|---|---|---|---|
| SVHN | with TS Temperature | 0.72±0.26 1.82 | 1.35±0.11▲ 1.11 | 1.22±0.08▲ 1.12 | 0.80±0.22▲ 1.10 |
| | with HB | 0.68±0.22 | 0.70±0.21▲ | 0.73±0.20▲ | 0.96±0.14▲ |
| CIFAR-10 | with TS Temperature | 0.95±0.19 2.51 | 2.54±0.11▲ 0.96 | 2.71±0.36▲ 0.95 | 1.39±0.28▲ 0.76 |
| | with HB | 0.74±0.15 | 0.94±0.21▲ | 1.16±0.54▲ | 1.65±0.31▲ |
| CIFAR-100 | with TS Temperature | 1.35±0.19 2.19 | 1.37±0.27▲ 1.04 | 3.92±0.21▲ 1.24 | 2.14±0.42▲ 0.97 |
| | with HB | 1.27±0.27 | 2.01±0.22▲ | 1.56±0.44▲ | 1.83±0.30▲ |
| 20 Newsgroups | with TS Temperature | 3.11±0.33 4.18 | 5.22±0.60▲ 1.06 | 2.71±0.25▼ 1.48 | 3.77±0.41▲ 0.89 |
| | with HB | 2.52±0.47 | 2.67±0.82▲ | 2.61±0.95▲ | 3.16±0.97▲ |

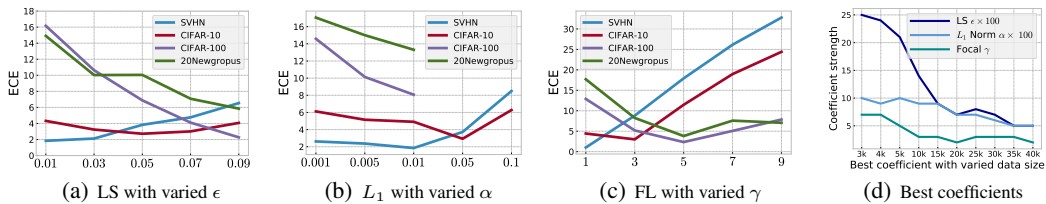

(a) LS with varied $\epsilon$     (b) $L_1$ with varied $\alpha$     (c) FL with varied $\gamma$     (d) Best coefficients

Figure 1: (a-c): ECE (%) with $M = 15$ of regularization methods with controlled regularization strength. (d): Best coefficients of regularization methods with respect to ECE with controlled training data size.

accuracy is relatively lower, the regularization methods need larger coefficients for better calibration. Based on this observation, we conduct another experiment for investigating the correlation between the regularization coefficients and accuracy. We learn networks on CIFAR-10 by controlling training data size, which leads to varied predictive accuracies, and choose the best coefficient for each case. Here, $\epsilon$, $\alpha$ and $\gamma$ are chosen from $\{0.01, 0.02, ..., 0.25\}$, $\{0.01, 0.02, ..., 0.1\}$ and $\{1, 3, 5, 7, 9\}$ respectively. Figure 2(d) shows that with the increase of training data size, which results in increase of predictive accuracy, the best coefficients of the regularization methods keep decreasing.

**Reliability Diagram** We use reliability diagram to visually represent the gap between predictive confidence and accuracy of each method. Due to the space limitation, here we only present the diagrams of CIFAR-10, and the rest figures are presented in Appendix. We can see that these visual results are similar with the comparison results of ECE reported in Table 2. Although the gap between confidence and accuracy is large when using the standard CE loss, it can be significantly diminished after using TS. However, the improvements of TS for the regularization methods are not obvious. Most importantly, no matter whether TS is used or not, the regularization methods suffer from overconfidence on samples which have high predictive uncertainty, especially on label smoothing and $L_1$ Norm, which contradicts the traditional view. Combining with the observation in Figure 2(d), it is indicated that the regularization methods work by simply aligning the average predictive confidence of the whole dataset to the accuracy with some specific regularization strengths, and does not produce fine-grained calibration with respect to the difference of samples.

**Impact of Validation Size** We also wonder how does the validation data size impact the post-hoc calibration. Figure 3 shows the ECE results with controlled validation data size. We can see that quite low ECE can be obtained with only a small size of validation data when using TS, which offers high efficiency for practice development. Relatively, HB needs more validation samples to obtain better calibration performance. Nevertheless, the standard CE loss stably achieves better calibration across varied validation data size with both TS and HB.

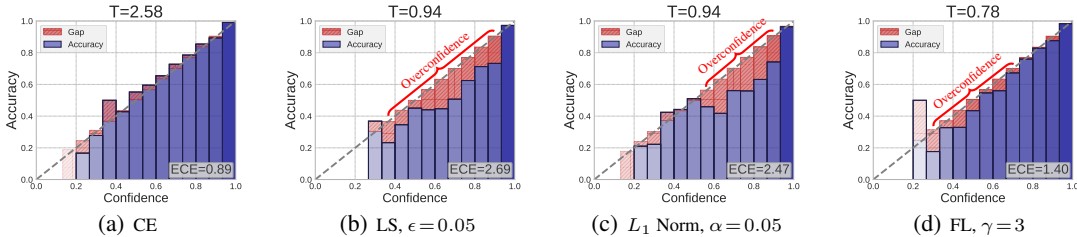

(a) CE      (b) LS, $\epsilon = 0.05$      (c) $L_1$ Norm, $\alpha = 0.05$      (d) FL, $\gamma = 3$

Figure 2: Reliability diagrams of each methods (after TS calibration) on CIFAR-10. The results are chosen from one of the 5 random runs of Table 1. Darker color of bars indicates that more samples are assigned with the corresponding confidence intervals.

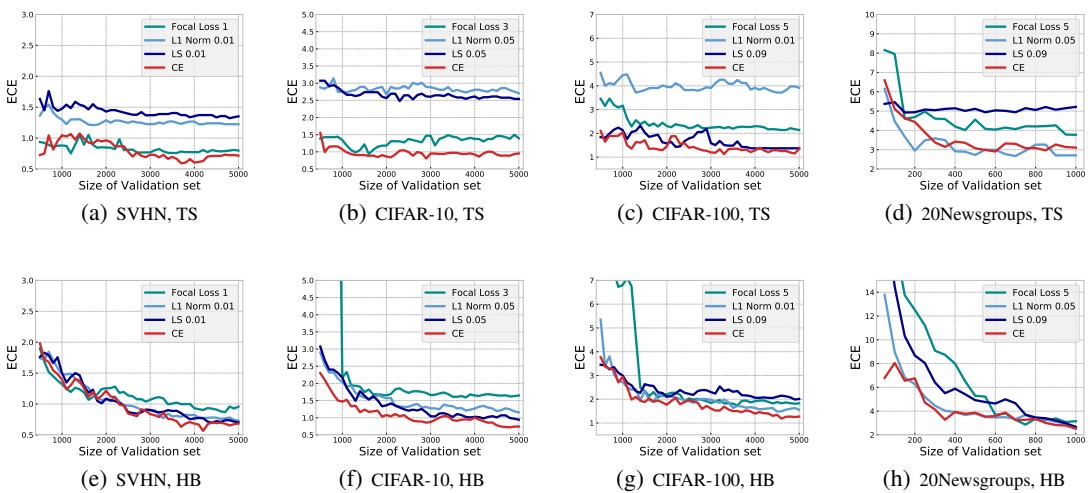

(a) SVHN, TS      (b) CIFAR-10, TS      (c) CIFAR-100, TS      (d) 20Newsgroups, TS

(e) SVHN, HB      (f) CIFAR-10, HB      (g) CIFAR-100, HB      (h) 20Newsgroups, HB

Figure 3: ECE (%) (after post-hoc calibration) with of regularization methods with controlled validation data size.

## 5 From *Calibrated* to *Calibratable*: A Closer Look

The results reported in above section show the degradation of regularization methods when integrating them with post-hoc calibration methods, which indicates that though regularization helps DNNs obtain well-calibrated predictions, it makes these predictions worse calibratable. In this Section, we further investigate this phenomenon by a series of illustrative experiments. We firstly attempt to empirically understand the reason of the calibration degradation from the view of information loss. Then, we propose an inverse form of focal loss to give a closer look at the correlation between the loss functions used in training and the calibration performance. The implementation details used in this section are also the same with those in Section 3.

### 5.1 Information Loss of Regularized Models

**ECE among Epochs** We start by investigating the ECEs of temperature-scaled outputs over epochs during model training. To avoid the impact of the bias of validation data, we directly search the best temperature on test data in each training epoch. We denote the corresponding ECE with this searched temperature as *optimal ECE*, which is the lower bound of ECE with temperatures searched on validation data. Figure 4 shows the curves of optimal ECE during epochs using label smoothing with different smoothing coefficients. We can observe that the optimal ECE rises after some learning epochs: On SVHN and CIFAR-10, it starts to significantly rise around the 10th epoch, and on CIFAR-100 and 20Newsgroups, it tends to rise after 100 and 50 epochs, where the learning rate drops by a factor of 10. Another observation is that larger smoothing strength $\epsilon$ results in worse calibration performance and more remarkable (also earlier) ECE rising. According to the *memorization effect* [35], DNNs usually learn easy samples at the early stage of training and tend to fit the hard ones later.

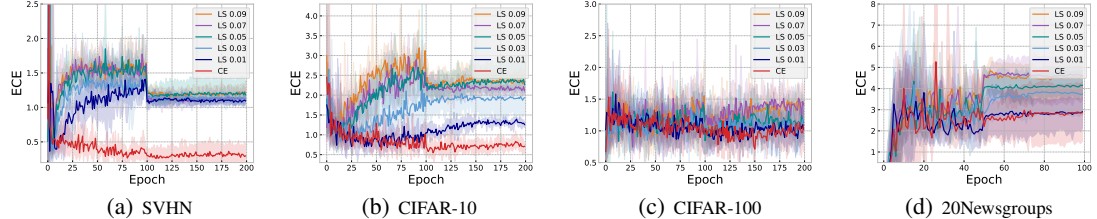

Figure 4: Curves of optimal ECE (%) during learning epochs using label smoothing with different coefficients. Dark colors show the mean results of 5 random runs and light colors show the ranges between minimal and maximum results of 5 runs.

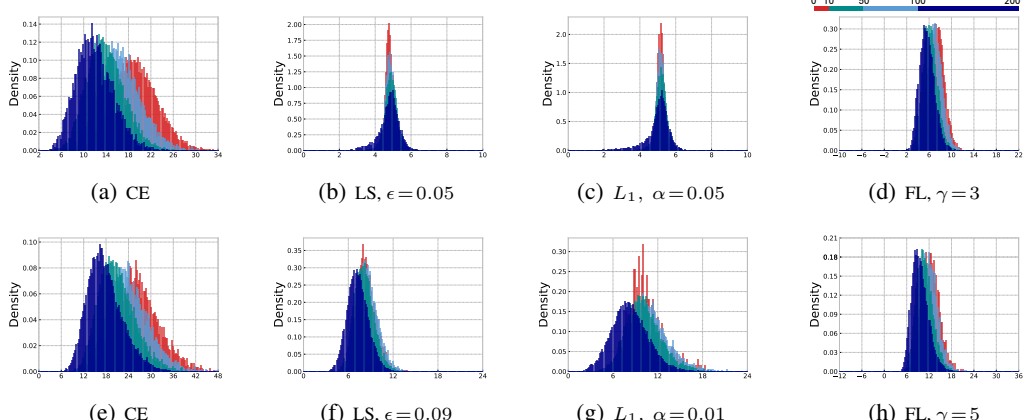

Figure 5: Histograms of maximum logits produced from models trained with different methods. Different colors of bars represent distributions of samples with different learned epochs. The first row and second row are results of CIFAR-10 and CIFAR-100 respectively.

Therefore, a simple conjecture for the ECE rising is that after some learning epochs, DNNs start to fit hard samples, at the same time the regularizer would penalize the confidences of easy samples, which makes the predictive confidences of those easy and hard samples difficult to be distinguished. Similar phenomenon is also observed when using $L_1$ Norm (see Appendix Figure A(a-d)) except 20Newsgroups dataset, on which large norm coefficient will hurt the calibration. For focal loss (see Appendix Figure A(e-h)), the optimal ECEs trained with large regularization strengths keep high without the remarkable rising.

**Histogram of Logits** We use histograms to visualize what the patterns of model outputs learned with different methods look like. Before that, we define *learned epoch*[2] of an individual training sample as the epoch, since which the sample can be correctly classified till the final learning epoch. As we mentioned above, DNNs usually learn hard samples after easy ones, hence the learned epoch of a sample can be used to indicate its corresponding hardness degree to be learned. Based on this, we want to investigate if the samples with different hardness degrees can be distinguished by model itself after training. To this end, we record the learned epochs of all training samples of CIFAR-10 and CIFAR-100 during training and statistic the distributions of their maximum logit outputs (i.e. $\max_i g_i^\theta(\boldsymbol{x})$). As shown in Figure 5, the logits of models trained with the standard CE loss cover much larger ranges, and the regularization methods compress the distributions too tight without distinction between samples with different learned epochs, especially in label smoothing and $L_1$ Norm. This visual observation further confirms that the regularization of DNNs works by only penalizing the confidence of the whole dataset to a low level with a specific regularization strength. This will result in loss of the important information about the hardness of samples as an undesirable side effects, and compress the room of potential improvement by post-hoc calibration. On the contrary, models trained

---

[2]This concept is inspired by the related work [30], in which the authors qualify a sample as *unforgettable* if it is learned at some epoch and experience no forgetting events during training.

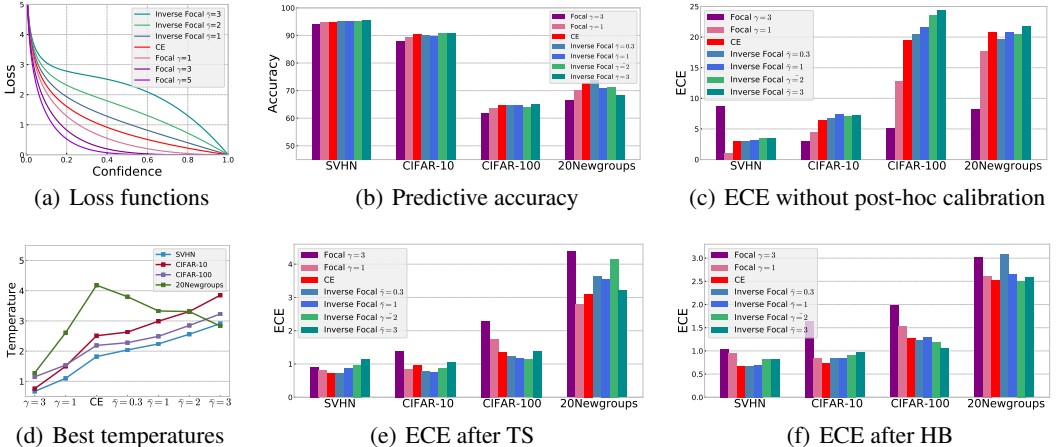

| (a) Loss functions | (b) Predictive accuracy | (c) ECE without post-hoc calibration |
|---|---|---|
| (d) Best temperatures | (e) ECE after TS | (f) ECE after HB |

Figure 6: (a): Visual representation of focal loss, CE loss and inverse focal loss. (b): Predictive accuracies (%) of different methods. (c): ECEs (%) with $M = 15$ of different methods without post-hoc calibration. (d): Searched temperatures on validation data. (e-f): ECEs (%) with $M = 15$ of different methods with the help of post-hoc calibration.

with the standard CE loss manage to preserve this information to a certain extent during training, hence achieve better results after post-hoc calibration.

## 5.2 Is Cross-Entropy the Best for Calibration?

Based on our experimental findings, one natural question is that can we design some loss functions in the opposite direction of these regularization methods to further improve the calibration? For label smoothing and $L_p$ Norm, we can simply set the regularization coefficients of these methods as negative values. However, we empirically found this will cause extremely low predictive accuracies even failures of training using only very small weighting coefficients. Fortunately, we can design an inverse version of focal loss without prediction degradation by mimicking the original focal loss[3]. Recall the form of focal loss, we see that it works by assigning larger weights to the samples with smaller confidences. This makes the optimizer pay more attention to those hard samples when updating model parameters. Actually, this weighting scheme also implicitly exists in the standard CE loss, and this can be expressed by the gradients of CE loss function w.r.t. model parameters $\theta$:

$$\frac{\partial \mathcal{L}_{ce}(y, \boldsymbol{f}^\theta(\boldsymbol{x}))}{\partial \theta} = -\frac{1}{f_y^\theta(\boldsymbol{x})} \nabla_\theta f_y^\theta(\boldsymbol{x}) \tag{3}$$

where the factor term $\frac{1}{f_y^\theta(\boldsymbol{x})}$ indicates that samples with smaller confidences are weighted larger in gradient calculation. Opposite to the principle of focal loss, we propose *inverse focal loss* as follows:

$$\mathcal{L}_{\bar{f}} = -(1 + f_y^\theta)^{\bar{\gamma}} \log f_y^\theta \tag{4}$$

By a simple modification on the weighting term of original focal loss, the inverse focal loss assigns larger weights to the samples with larger output confidences. Similar with original focal loss, the choice of coefficient $\bar{\gamma}$ has a huge impact on the property of this loss. In Figure 6(a), we plot the curves of inverse focal loss with varied $\bar{\gamma}$ and also plot the standard CE loss and focal loss for comparison. We can see that different from the original focal loss, the curves of inverse focal loss are steeper when confidence is large, and larger $\bar{\gamma}$ gives steeper curves.

We conduct another experiment to evaluate the inverse focal loss, and also investigate what will happen when we increase its coefficient $\bar{\gamma}$. Figure 6(c) shows the ECE results without post-hoc calibration. The ECEs of inverse focal loss are larger than CE and focal loss in most cases. This is consistent to our expectation since inverse focal loss aggravates the overconfidence issue of DNNs by weighting larger on the easy samples. Figure 6(e) and 6(f) show the ECE results with the help of

---

[3]As the reviewer suggested, there already exists an "inverse focal loss" in the literature, which was introduced in a totally different context and using a different mathematical expression, but similar motivation [15].

post-hoc calibration. When using HB, the ECEs of inverse focal loss are worse than CE on SVHN and CIFAR-10, while better than CE on CIFAR-100. Generally speaking, there is no clear trend when we increase $\bar{\gamma}$. More interesting results appear when using TS: (1) On CIFAR-10 and CIFAR-100, the ECE results of inverse focal loss are better than that of the standard CE loss; and (2) there is a *descend-then-ascend* trend from focal loss with $\gamma = 3$ to inverse focal loss with $\bar{\gamma} = 3$. From these observations, we may say that the best loss function for calibration is varied across different tasks according to the characteristics of datasets. On SVHN, which is a relatively easy dataset, standard CE loss yields pretty good results; on CIFAR-10 and CIFAR-100, which is more complex and difficult, the best results are obtained using inverse focal loss; on 20Newsgroups, which has fewest training samples among four datasets, the best result is obtained when using focal loss with $\gamma = 1$. The searched temperatures when using TS are presented in Figure 6(d). The increasing of best temperatures indicates that the overconfidence problem is severer when using inverse focal loss with larger $\bar{\gamma}$. The predictive accuracies are presented in Figure 6(a). As is shown that inverse focal loss yields highly competitive results compared with the standard CE loss on SVHN, CIFAR-10 and CIFAR-100. On 20Newsgroups, when using large $\bar{\gamma}$, the predictive performance of inverse focal loss is worse than the CE loss.

## 6 Related Work

In machine learning, calibration has long been studied [25, 33, 34, 22], and many classical methods, like Platt Scaling [25] and Histogram Binning [33], have been proposed in the literature. In recent years, deep neural networks trained with commonly used CE loss, have been empirically found to predict poorly calibrated probabilities. The early researches for this problem focus on bayesian models [2, 16, 3, 1], which indirectly infer prediction uncertainty through weight uncertainties. But training and inferring the bayesian DNNs are computationally more expensive and conceptually more complicated than deterministic DNNs. Therefore, the uncertainty qualification of the non-bayesian models has always been an important topic, which also attracts a lot of researchers from the perspective of understanding DNNs.

Guo *et al.* [4] systematically investigated the miscalibration problem of the deterministic DNNs and empirically compared several conventional post-hoc calibration fixes. Two key findings are suggested in their paper: (1) Increasing model capacity and regularization strength negatively affect the calibration. (2) Simple post-hoc methods like TS [25] and HB [33] can reduce the calibration error to a quite low level. Following their work, there is a surge of research that proposed new post-hoc calibration methods [10, 27, 5, 11, 24, 36, 26]. Different from post-hoc calibration methods, another line of research aims to learn calibrated networks during training by modifying the training process [29, 12, 8]. Thulasidasan *et al.* [29] found that DNNs trained with mixup are significantly better calibrated than DNNs trained in the regular fashion. Kumar *et al.* [12] proposed a RKHS kernel based measure of calibration that is efficiently trainable alongside the standard CE loss, which can minimize an explicit calibration error during training. Krishnan and Tichoo [8] introduced a differentiable accuracy versus uncertainty calibration loss function that allows a model to learn to provide well-calibrated uncertainties, in addition to improved accuracy. Recently, inspired by the findings in [4], several studies were proposed to leverage the regularization of DNNs to improve calibration performance during training [19, 18, 7]. As we described in Section 3, these implicit or explicit regularization techniques can improve calibration by penalizing the predictive confidences of DNNs. It is worth nothing that besides the studies on improving calibration performance, there are also several studies that focus on the measure of calibration performance [23, 31, 32, 5].

## 7 Conclusion

In this work, we investigate the uncertainty calibration problem of DNNs by a series of experiments. The empirical study shows that despite the predictions obtained from the regularized models are better calibrated, worse results would be obtained if we employ post-hoc calibration methods on these regularized models. Extended experiments demonstrate that the regularized DNNs will lose the important information about the hardness of samples, which results in the harm of post-hoc calibration. Based on the experimental observations, we design a new loss function in the opposite direction of previous regularization methods, and empirically show the superiority of this loss in calibration with the help of post-hoc methods, even though it causes severer overconfidence issue in

the main training phase. Our findings suggest that overconfidence of DNNs is not the nightmare in model calibration and point out a new direction to improve the calibration performance of DNNs by considering main training and post-hoc calibration as a unified framework. Moreover, the study of the phenomena of deep learning uncertainty under distribution shift is very interesting as one of the future work, since the behaviour with distribution shifts might be most important in practice.

## 8    Acknowledgments

The authors wish to thank the anonymous reviewers for their helpful comments and suggestions. This work was supported by the National Science Foundation of China (62176055), the Postgraduate Research & Practice Innovation Program of Jiangsu Province (KYCX21_0151) and the China University S&T Innovation Plan Guided by the Ministry of Education. We thank the Big Data Center of Southeast University for providing the facility support on the numerical calculations in this paper.

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
