# Appendix

## A  About Equation (1)

As we discussed in Section 3, label smoothing and focal loss are equivalent to the standard CE loss with an additional maximum-entropy regularizer (see in Equation (1) and (2) in the main text). The proof of Equation (2) can be found in the corresponding paper [4]. We did not find the proof of Equation (1) in related work, thus here we give its simple proof for the readability:

$$\mathcal{L}_{ce}(\widetilde{\boldsymbol{y}}, \boldsymbol{f}^\theta) = -\sum_{k=1}^K \widetilde{y}_k \log f_k^\theta = -\sum_{k=1}^K [(1-\epsilon)y_k + \frac{\epsilon}{K}] \log f_k^\theta$$

$$= (1-\epsilon)[-\sum_{k=1}^K y_k \log f_k^\theta] + \epsilon[-\frac{1}{K}\sum_{k=1}^K \log f_k^\theta]$$

$$= (1-\epsilon)\mathcal{L}_{ce}(\boldsymbol{y}, \boldsymbol{f}^\theta) + \epsilon\mathcal{L}_{ce}(\boldsymbol{u}, \boldsymbol{f}^\theta)$$

## B  Experimental Settings

**Datasets**  The experiments in main text are based on four popular datasets: SVHN [5], CIFAR-10, CIFAR-100 [2] and 20Newsgropus [3]. SVHN is an image dataset which consists of 32×32 colored images of 0∼9 digits. CIFAR-10 and CIFAR-100 consist of 32×32 colored natural images arranged in 10 and 100 classes, respectively. 20Newsgropus is a collection of news documents, partitioned nearly evenly across 20 different newsgroups. In our experiments, we split each dataset into train/validation/test sets as following ratios: 68257/5000/26032 for SVHN, 45000/5000/10000 for CIFAR-10/100 and 13828/1000/4000 for 20Newsgroups.

**Implement Details**  The experiments in main text use ResNet-32 and a 8-layer 1D-CNN as the base models. For 20Newsgroups, we use the GloVe word embedding [7] for text representation before the 1D-CNN model and set the embedding dimension as 100. The structure of ResNet can be found in the original paper [1], and the structure of the used 1D-CNN is as follows:

[Conv1D(100,128,3) ==> Relu() ==> MaxPool1D(3)] × 4 ==> Conv1D(100,128,3) ==> Relu() ==> MaxPool1D(9) ==> Linear(128,100) ==> Relu() ==> Linear(100,50) ==> Relu() ==> Linear(50,20) ==> SoftMax()

Conv1D($a$,$b$,$c$) denotes the 1-dimensional convolution process with $a$ input channels, $b$ output channels and kernel size as $c$. MaxPool1D($c$) denotes the 1-dimensional max pooling process with kernel size as $c$. Linear($a$,$b$) denotes the fully connectedly layer with $a$ input units and $b$ output units.

We use SGD as the opimizer with a momentum of 0.9, a weight decay of 1e-4. For SVHN and CIFAR-10/100, we set batch size as 512, the initial learning rate as 0.1 and divide it by a factor of 10 after 100 epochs and 150 epochs respectively. For 20Newsgroups, we set batch size as 128, the initial learning rate as 0.05 and divide it by a factor of 10 after 50 epochs, 100 epochs and 150 epochs respectively. The implementation is based on PyTorch [6] and the experiments were carried out with NVIDIA Tesla V100 GPU.

## C  Optimal ECE Curves

Figure A shows the optimal ECE curves of $L_1$ Norm and focal loss with different coefficients. The curves of $L_1$ Norm are very similar with those of label smoothing, where the optimal ECE rises after some learning epochs and using larger coefficient results in worse calibration performance and more remarkable ECE rising. For focal loss, the optimal ECEs trained with large coefficients keep high without the remarkable rising.

## D  Reliability Diagrams

Figure B-E show the reliability diagrams of each method with and without post-hoc calibration. We can see that these visual results are similar with the comparison results of ECE reported in Table 2 in the main text. Although the gaps between confidences and accuracies of models trained with the CE

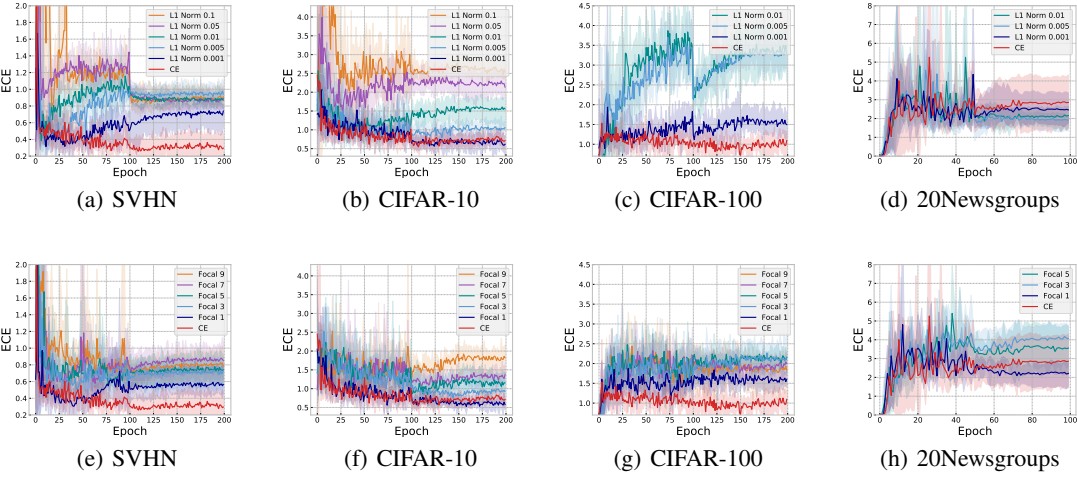

Figure A: Curves of optimal ECE (%) during learning epochs using $L_1$ Norm and focal loss with different coefficients. Dark colors show the mean results of 5 random runs and light colors show the ranges between minimal and maximum results of 5 runs.

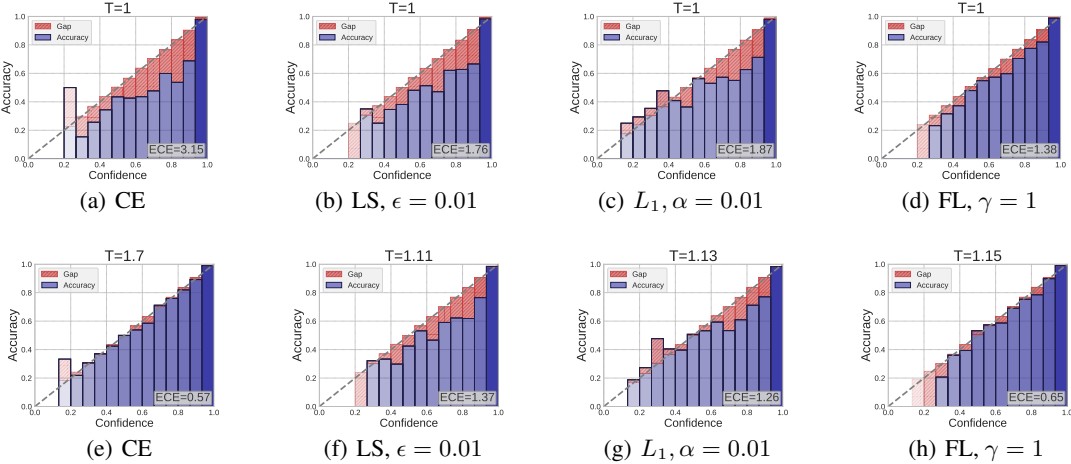

Figure B: Reliability diagrams of each methods on SVHN. The first row presents the results without TS and the second row shows the results after TS. The results are chosen from one of the 5 random runs of Table 1 in the main text.

loss are very large on all four datasets, they can be significantly diminished after using TS. However, the improvements of TS for the other three methods are not obvious.

## E  Comparison Results with ResNet-110

The experiments on image datasets in main text are conducted based on ResNet-32. Here, we also report the experimental results on CIFAR-10/100 using a deeper model ResNet-110, in which we use the same experimental settings with those described in Section A.2. The comparison results are reported in Table A. The results are similar with Section 3 and Section 4 in the main text: (1) Regularization-based methods significantly degrade the ECE on all datasets compared with the standard CE loss; (2) the standard CE loss achieves the best calibration performance on most cases when post-hoc calibration methods are used; (3) The searched temperatures of models trained with the CE loss are significantly higher than those of other losses, which indicates that the CE loss causes higher predictive confidences. Also note that the searched temperatures of the CE loss on ResNet-110

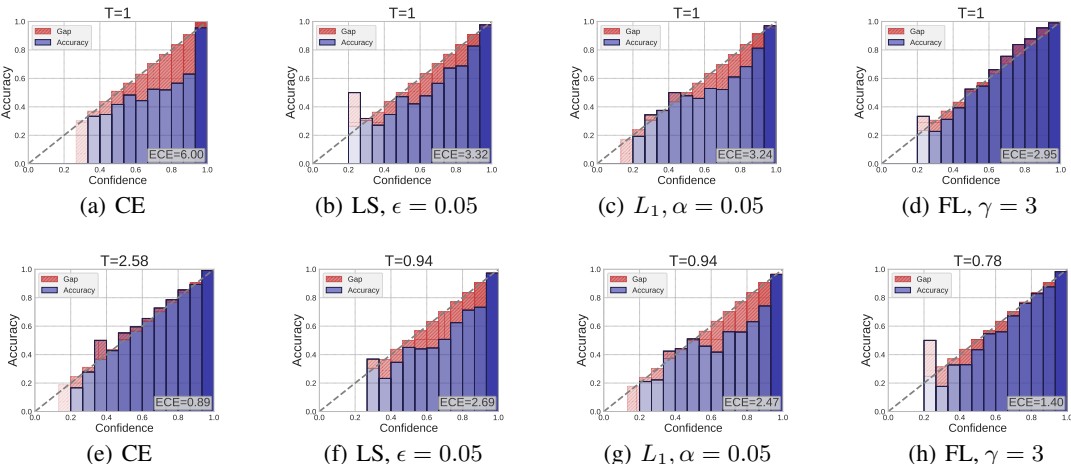

Figure C: Reliability diagrams of each methods on CIFAR-10. The first row presents the results without TS and the second row shows the results after TS.

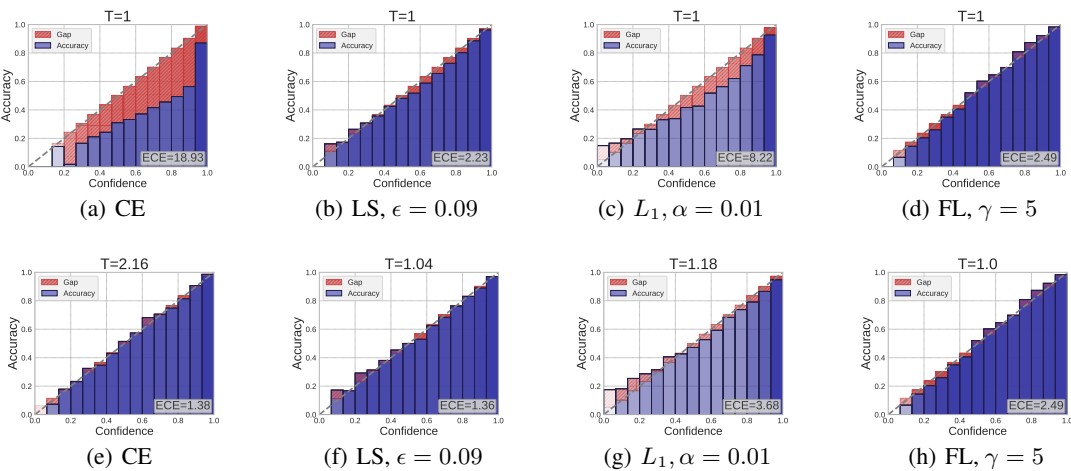

Figure D: Reliability diagrams of each methods on CIFAR-100. The first row presents the results without TS and the second row shows the results after TS.

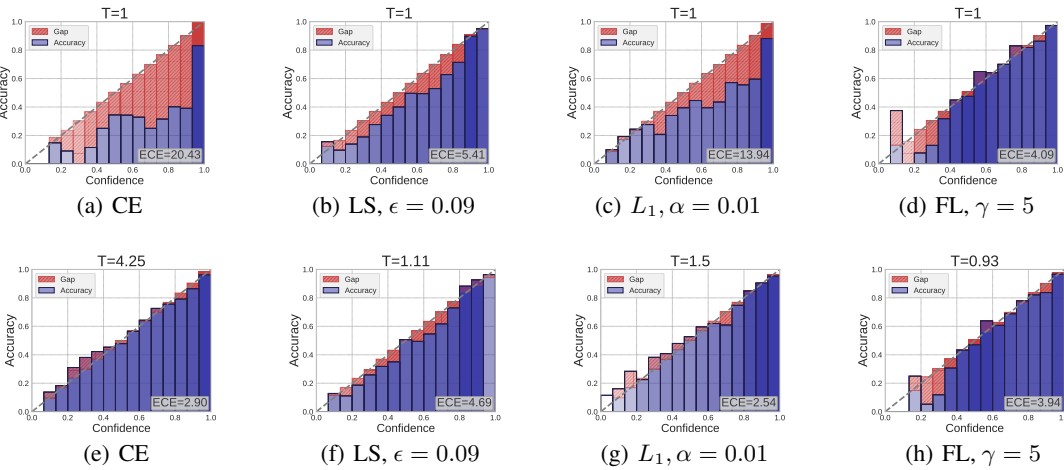

Figure E: Reliability diagrams of each methods on 20Newsgroups. The first row presents the results without TS and the second row shows the results after TS.

Table A: Comparison results (mean±std) of ECE (%) with $M = 15$ using ResNet-110 over 5 random runs. The values with underline in first row represent the best coefficients chose for each regularization method on each dataset according to the ECE on test data. ▲/▲ and ▼/▼ indicate that the average ECE of regularization methods are higher and lower than standard CE, where ▲ and ▼ are based on two-sample t-test at 0.05 significance level.

| | | Cross-Entropy | Label Smoothing 0.09/0.09 | $L_1$ Norm 0.05/0.005 | Focal Loss 3/7 |
|---|---|---|---|---|---|
| CIFAR-10 | w/o post | 8.06±0.66 | 3.78±0.87 | 3.90±0.37 | 2.73±0.51 |
| | with TS Temperature | 0.77±0.15 2.94 | 2.68±0.36▲ 0.96 | 2.89±0.60▲ 1.03 | 1.11±0.35▲ 0.85 |
| | with HB | 0.99±0.42 | 0.86±0.17▼ | 1.24±0.31▲ | 1.03±0.12▲ |
| CIFAR-100 | w/o post | 23.68±1.09 | 4.74±0.76 | 15.87±1.99 | 3.12±0.29 |
| | with TS Temperature | 1.29±0.35 2.62 | 1.89±0.45▲ 1.09 | 2.94±0.58▲ 1.38 | 3.09±0.23▲ 0.98 |
| | with HB | 1.21±0.69 | 1.72±0.31▲ | 1.62±0.35▲ | 1.65±0.30▲ |

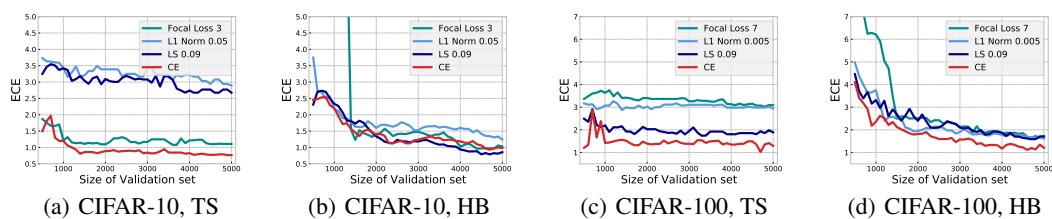

(a) CIFAR-10, TS      (b) CIFAR-10, HB      (c) CIFAR-100, TS      (d) CIFAR-100, HB

Figure F: ECE (%) (after post-hoc calibration) of regularization methods using ResNet-110 with controlled validation data size.

are larger than those on ResNet-32. The coefficient $\alpha$ is chosen from {0.001, 0.005} on CIFAR-100 since $L_1$ Norm fails on it when $\alpha \geq 0.01$. We also report the ECE results with controlled validation data size. As we can see that the standard CE loss stably achieves better calibration across varied validation data size with both TS and HB.

## F  Full Comparison Results

Table B-G show the full comparison results of the predictive accuracy and the ECE. We can see that for each regularization method, the ECE results before post-hoc calibration are lower than the standard CE loss in most cases. After post-hoc calibration, as the coefficients of regularization methods increase, their calibration results usually get worse and worse especially when using TS.

## G  Broader Impact

A reliable predictor is expected to be accurate when it is confident about its predictions and indicate high uncertainty when it is likely to be inaccurate. This uncertainty quantification ability is very important in many real-world machine learning applications. As we discussed in related work section in the main text, there are two main lines of research in this area. The first one aims to learn calibrated predictions during training and the second one aims to calibrate the predictions after main training using post-hoc methods. Our empirical study shows that some of methods in the first line cannot achieve better calibration performance than the standard CE loss when integrating them with post-hoc methods.

By a series of experiments, our work gives a closer look at the calibration of DNNs and point out that we should consider main training and post-hoc calibration as a unified framework, and maybe we should pay more attention to obtain calibratable predictions rather than calibrated ones in DNN training. Interestingly, our work finds that the standard CE loss actually has the quality of preserving the information about the hardness of samples, which can make the learned models obtain well calibratable predictions after training. Since this work is mostly on the empirical study of the calibration problem in DNNs, we do not see any direct negative impacts on society, and we

Table B: Full comparison results (mean±std) of the predictive accuracy (%) and ECE (%) with $M = 15$ over 5 random runs on SVHN.

|  |  | Accuracy | ECE w/o post | ECE with TS | ECE with HB |
|---|---|---|---|---|---|
| Cross-Entropy | — | 95.00±0.27 | 3.03±0.16 | 0.72±0.26 | 0.68±0.22 |
| Label Smoothing | $\epsilon = 0.01$ | 95.21±0.23 | 1.84±0.19▼ | 1.35±0.11▲ | 0.70±0.21▲ |
|  | $\epsilon = 0.03$ | 95.19±0.17 | 2.12±0.12▼ | 1.54±0.16▲ | 0.82±0.31▲ |
|  | $\epsilon = 0.05$ | 95.18±0.11 | 3.83±0.08▲ | 1.42±0.17▲ | 0.76±0.13▲ |
|  | $\epsilon = 0.07$ | 95.22±0.14 | 4.76±0.06▲ | 1.42±0.06▲ | 0.81±0.15▲ |
|  | $\epsilon = 0.09$ | 95.19±0.19 | 6.54±0.11▲ | 1.54±0.09▲ | 0.85±0.15▲ |
| $L_1$ Norm | $\alpha = 0.001$ | 95.54±0.04 | 2.62±0.12▼ | 0.87±0.15▲ | 0.63±0.22▼ |
|  | $\alpha = 0.005$ | 95.22±0.31 | 2.37±0.24▼ | 1.12±0.11▲ | 0.61±0.16▼ |
|  | $\alpha = 0.01$ | 95.29±0.13 | 1.85±0.04▼ | 1.22±0.08▲ | 0.73±0.20▲ |
|  | $\alpha = 0.05$ | 95.55±0.04 | 3.72±0.12▲ | 1.24±0.16▲ | 0.95±0.20▲ |
|  | $\alpha = 0.1$ | 95.59±0.10 | 8.50±0.10▲ | 1.27±0.11▲ | 0.81±0.05▲ |
| Focal Loss | $\gamma = 1$ | 94.77±0.19 | 1.01±0.21▼ | 0.80±0.22▲ | 0.96±0.14▲ |
|  | $\gamma = 3$ | 94.06±0.49 | 8.80±0.60▲ | 0.91±0.14▲ | 1.04±0.40▲ |
|  | $\gamma = 5$ | 93.34±0.55 | 17.91±1.38▲ | 0.98±0.23▲ | 1.62±0.52▲ |
|  | $\gamma = 7$ | 92.56±0.52 | 26.14±0.62▲ | 1.03±0.21▲ | 1.72±1.26▲ |
|  | $\gamma = 9$ | 92.44±0.32 | 32.78±0.69▲ | 0.84±0.13▲ | 3.77±4.57▲ |

Table C: Full comparison results (mean±std) of the predictive accuracy (%) and ECE (%) with $M = 15$ over 5 random runs on CIFAR-10.

|  |  | Accuracy | ECE w/o post | ECE with TS | ECE with HB |
|---|---|---|---|---|---|
| Cross-Entropy | — | 90.46±0.23 | 6.43±0.22 | 0.95±0.19 | 0.74±0.15 |
| Label Smoothing | $\epsilon = 0.01$ | 90.42±0.62 | 4.32±0.35▼ | 1.47±0.27▲ | 1.04±0.25▲ |
|  | $\epsilon = 0.03$ | 90.33±0.27 | 3.24±0.21▼ | 2.50±0.09▲ | 0.91±0.25▲ |
|  | $\epsilon = 0.05$ | 90.09±0.41 | 2.72±0.32▼ | 2.54±0.11▲ | 0.94±0.21▲ |
|  | $\epsilon = 0.07$ | 90.15±0.55 | 3.01±0.12▼ | 2.63±0.19▲ | 0.90±0.19▲ |
|  | $\epsilon = 0.09$ | 89.72±0.44 | 4.07±0.35▼ | 3.00±0.08▲ | 1.07±0.18▲ |
| $L_1$ Norm | $\alpha = 0.001$ | 89.97±0.79 | 6.12±0.49▼ | 0.91±0.17▼ | 0.78±0.20▲ |
|  | $\alpha = 0.005$ | 90.55±0.63 | 5.15±0.38▼ | 1.29±0.37▲ | 0.84±0.15▲ |
|  | $\alpha = 0.01$ | 90.44±0.42 | 4.91±0.40▼ | 1.90±0.54▲ | 1.11±0.33▲ |
|  | $\alpha = 0.05$ | 90.06±0.59 | 2.93±0.39▼ | 2.71±0.36▲ | 1.16±0.54▲ |
|  | $\alpha = 0.1$ | 90.03±0.19 | 6.28±0.21▼ | 2.78±0.35▲ | 1.18±0.32▲ |
| Focal Loss | $\gamma = 1$ | 89.29±0.34 | 4.43±0.22▼ | 0.84±0.28▼ | 0.84±0.15▲ |
|  | $\gamma = 3$ | 87.84±0.17 | 3.00±0.26▼ | 1.39±0.28▲ | 1.65±0.31▲ |
|  | $\gamma = 5$ | 85.95±0.42 | 11.45±0.19▲ | 1.18±0.14▲ | 1.39±0.46▲ |
|  | $\gamma = 7$ | 84.07±0.63 | 18.96±0.37▲ | 1.47±0.39▲ | 6.71±3.38▲ |
|  | $\gamma = 9$ | 81.50±0.68 | 24.40±0.75▲ | 1.42±0.13▲ | 5.78±2.79▲ |

believe that more attention should be paid on the uncertainty quantification of DNNs to help safety and fairness in AI.

## H  Additional Results During Review Phase

We have conducted some new results during review phase, e.g., ECEs of models trained with smaller batch size (which results in higher accuracy) and results on other calibration metrics. Here we present

Table D: Full comparison results (mean±std) of the predictive accuracy (%) and ECE (%) with $M = 15$ over 5 random runs on CIFAR-100.

| | | Accuracy | ECE w/o post | ECE with TS | ECE with HB |
|---|---|---|---|---|---|
| Cross-Entropy | — | 64.64±0.43 | 19.53±0.36 | 1.35±0.19 | 1.27±0.27 |
| Label Smoothing | $\epsilon = 0.01$ | 63.85±0.32 | 16.16±0.32▼ | 1.31±0.26▼ | 1.55±0.28▲ |
| | $\epsilon = 0.03$ | 64.29±0.20 | 10.63±0.18▼ | 1.40±0.26▲ | 1.71±0.68▲ |
| | $\epsilon = 0.05$ | 64.11±0.36 | 6.88±0.20▼ | 1.39±0.20▲ | 1.36±0.15▲ |
| | $\epsilon = 0.07$ | 64.28±0.30 | 4.11±0.37▼ | 1.55±0.35▲ | 1.55±0.39▲ |
| | $\epsilon = 0.09$ | 63.73±0.67 | 2.27±0.48▼ | 1.37±0.27▲ | 2.01±0.22▲ |
| $L_1$ Norm | $\alpha = 0.001$ | 64.75±0.20 | 14.59±0.23▼ | 1.68±0.46▲ | 1.59±0.41▲ |
| | $\alpha = 0.005$ | 63.74±0.52 | 10.14±0.40▼ | 3.36±0.16▲ | 1.50±0.18▲ |
| | $\alpha = 0.01$ | 63.07±0.29 | 8.07±0.44▼ | 3.92±0.21▲ | 1.56±0.44▲ |
| Focal Loss | $\gamma = 1$ | 63.64±0.40 | 12.89±0.48▼ | 1.75±0.43▲ | 1.53±0.26▲ |
| | $\gamma = 3$ | 61.91±0.20 | 5.19±0.36▼ | 2.30±0.22▲ | 1.98±0.24▲ |
| | $\gamma = 5$ | 60.36±0.44 | 2.34±0.35▼ | 2.14±0.42▲ | 1.83±0.30▲ |
| | $\gamma = 7$ | 59.54±0.39 | 5.07±0.44▼ | 2.28±0.47▲ | 1.55±0.38▲ |
| | $\gamma = 9$ | 58.39±0.52 | 7.85±0.62▼ | 2.03±0.44▲ | 1.68±0.59▲ |

Table E: Full comparison results (mean±std) of the predictive accuracy (%) and ECE (%) with $M = 15$ over 5 random runs on 20Newsgroups.

| | | Accuracy | ECE w/o post | ECE with TS | ECE with HB |
|---|---|---|---|---|---|
| Cross-Entropy | — | 72.85±0.89 | 20.82±0.93 | 3.11±0.33 | 2.52±0.47 |
| Label Smoothing | $\epsilon = 0.01$ | 59.76±27.27 | 10.03±4.94▼ | 3.54±1.75▲ | 2.10±1.14▼ |
| | $\epsilon = 0.03$ | 73.49±1.01 | 10.05±0.55▼ | 4.75±0.65▲ | 3.23±1.12▲ |
| | $\epsilon = 0.05$ | 73.90±0.65 | 7.08±0.85▼ | 4.72±0.56▲ | 2.19±0.32▼ |
| | $\epsilon = 0.07$ | 72.81±0.26 | 5.85±0.64▼ | 5.22±0.60▲ | 2.67±0.82▲ |
| | $\epsilon = 0.09$ | 72.81±1.44 | 17.06±0.88▼ | 3.01±0.48▼ | 3.15±1.06▲ |
| $L_1$ Norm | $\alpha = 0.001$ | 73.62±0.71 | 15.01±0.41▼ | 2.47±0.34▼ | 3.79±0.73▲ |
| | $\alpha = 0.005$ | 73.61±0.80 | 13.31±0.56▼ | 2.71±0.25▼ | 2.61±0.95▲ |
| Focal Loss | $\gamma = 1$ | 70.23±1.93 | 17.69±0.72▼ | 2.80±0.76▼ | 2.61±0.56▲ |
| | $\gamma = 3$ | 66.51±1.26 | 8.24±1.21▼ | 4.38±0.27▲ | 3.01±0.62▲ |
| | $\gamma = 5$ | 59.17±1.81 | 3.82±0.51▼ | 3.77±0.41▲ | 3.16±0.97▲ |
| | $\gamma = 7$ | 54.81±1.37 | 7.58±0.86▼ | 2.93±0.86▼ | 6.00±2.42▲ |
| | $\gamma = 9$ | 49.94±3.74 | 7.06±1.92▼ | 3.08±0.74▼ | 6.31±2.34▲ |

two tables which report the ECE of more powerful models on CIFAR-10/100 and the results of other calibration metrics on CIFAR-10.

In the main experiment, we focused on the uncertainty problem hence didn't pay much attention on the accuracy. We find that simply changing the batch size from 512 to 128 can significantly improve the accuracy. For examples, On CIFAR-10, we achieve 92.24 and 94.48 using ResNet-32 and ResNet-110 respectively. On CIFAR-100, we achieve 67.23 and 75.32 using ResNet-32 and ResNet-110 respectively. Considering that 5000 training samples are discarded in training, we think these results are competitive. Moreover, as is shown in Table H, the ECEs (both before and after post-hoc calibration) of new model are very similar with the results of previous experiments (see Table ).

As the reviewer suggested, ECE is a biased measure and there are several other metrics which can be used for evaluating the calibration performance. Here we give some results of other metrics (including

Table F: Full comparison results (mean±std) of the predictive accuracy (%) and ECE (%) with $M = 15$ over 5 random runs on CIFAR-10 (using **ResNet-110**).

|  |  | Accuracy | ECE w/o post | ECE with TS | ECE with HB |
|---|---|---|---|---|---|
| Cross-Entropy | — | 89.10±0.87 | 8.06±0.66 | 0.77±0.15 | 0.99±0.42 |
| Label Smoothing | $\epsilon = 0.01$ | 87.95±0.39 | 7.28±0.81▼ | 2.35±0.37▲ | 0.98±0.24▼ |
|  | $\epsilon = 0.03$ | 88.66±0.67 | 5.76±0.25▼ | 2.59±0.15▲ | 1.36±0.17▲ |
|  | $\epsilon = 0.05$ | 87.91±1.21 | 4.07±0.91▼ | 3.26±0.47▲ | 0.90±0.28▼ |
|  | $\epsilon = 0.07$ | 86.69±0.81 | 4.08±0.54▼ | 3.02±0.37▲ | 1.00±0.15▲ |
|  | $\epsilon = 0.09$ | 87.09±2.15 | 3.78±0.87▼ | 2.68±0.36▲ | 0.86±0.17▼ |
| $L_1$ Norm | $\alpha = 0.001$ | 88.04±1.15 | 8.23±0.96▲ | 1.37±0.72▲ | 1.02±0.30▲ |
|  | $\alpha = 0.005$ | 87.85±1.20 | 8.53±1.05▲ | 2.54±0.44▲ | 0.84±0.18▼ |
|  | $\alpha = 0.01$ | 88.84±1.50 | 7.61±1.08▼ | 2.94±0.32▲ | 0.86±0.25▼ |
|  | $\alpha = 0.05$ | 88.46±1.35 | 3.90±0.37▼ | 2.89±0.60▲ | 1.24±0.31▲ |
|  | $\alpha = 0.1$ | 58.41±35.86 | 4.49±2.27▼ | 2.66±1.32▲ | 0.90±0.21▼ |
| Focal Loss | $\gamma = 1$ | 86.44±0.42 | 7.01±0.93▼ | 1.37±0.38▲ | 1.16±0.42▲ |
|  | $\gamma = 3$ | 84.05±1.58 | 2.73±0.51▼ | 1.11±0.35▲ | 1.03±0.12▲ |
|  | $\gamma = 5$ | 79.66±2.99 | 13.18±1.51▲ | 1.70±0.34▲ | 3.49±2.94▲ |
|  | $\gamma = 7$ | 78.68±2.29 | 19.44±0.85▲ | 2.30±0.38▲ | 3.55±1.01▲ |
|  | $\gamma = 9$ | 73.98±2.80 | 25.12±2.31▲ | 2.20±0.21▲ | 2.10±0.82▲ |

Table G: Full comparison results (mean±std) of the predictive accuracy (%) and ECE (%) with $M = 15$ over 5 random runs on CIFAR-100 (using **ResNet-110**).

|  |  | Accuracy | ECE w/o post | ECE with TS | ECE with HB |
|---|---|---|---|---|---|
| Cross-Entropy | — | 62.64±1.44 | 23.68±1.09 | 1.29±0.35 | 1.21±0.69 |
| Label Smoothing | $\epsilon = 0.01$ | 61.23±2.79 | 18.46±1.86▼ | 1.35±0.15▲ | 1.35±0.42▲ |
|  | $\epsilon = 0.03$ | 60.02±2.52 | 13.49±1.57▼ | 1.77±0.31▲ | 1.56±0.44▲ |
|  | $\epsilon = 0.05$ | 63.84±1.75 | 9.05±1.74▼ | 1.94±0.26▲ | 1.61±0.69▲ |
|  | $\epsilon = 0.07$ | 61.75±2.28 | 7.64±1.55▼ | 1.98±0.53▲ | 1.64±0.26▲ |
|  | $\epsilon = 0.09$ | 59.34±1.73 | 4.74±0.76▼ | 1.89±0.45▲ | 1.72±0.31▲ |
| $L_1$ Norm | $\alpha = 0.001$ | 63.18±0.39 | 18.02±0.57▼ | 2.29±0.31▲ | 1.82±0.44▲ |
|  | $\alpha = 0.005$ | 63.29±2.44 | 15.87±1.99▼ | 2.94±0.58▲ | 1.62±0.35▲ |
| Focal Loss | $\gamma = 1$ | 59.98±3.47 | 17.70±1.32▼ | 2.15±0.26▲ | 1.56±0.68▲ |
|  | $\gamma = 3$ | 58.49±1.27 | 8.73±0.44▼ | 2.91±0.30▲ | 1.57±0.14▲ |
|  | $\gamma = 5$ | 57.56±1.31 | 3.61±0.69▼ | 3.13±0.33▲ | 1.66±0.26▲ |
|  | $\gamma = 7$ | 52.93±1.32 | 3.12±0.29▼ | 3.09±0.23▲ | 1.65±0.30▲ |
|  | $\gamma = 9$ | 55.30±0.27 | 5.52±0.37▼ | 2.63±0.43▲ | 1.69±0.45▲ |

NLL, Brier, SCE and ACE) on CIFAR-10. As is shown in Table I, the results of these four metrics show similar phenomenon with the results of ECE.

You can also find some other results from the OpenReview page [1].

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

Table H: Comparison results (mean±std) of the predictive accuracy (%) and ECE (%) with $M = 15$ on CIFAR-10/100 over 5 random runs.

| | | Cross-Entropy | Label Smoothing 0.05 | $L_1$ Norm 0.05 | Focal Loss 3 |
|---|---|---|---|---|---|
| CIFAR-10 ResNet-32 | Accuracy | 92.24±0.17 | 91.64±0.66 | 92.13±0.22 | 90.71±1.23 |
| | ECE w/o post | 5.15±0.17 | 2.74±0.39 | 3.28±0.16 | 3.45±0.42 |
| | ECE with TS | 0.76±0.10 | 2.09±0.22▲ | 2.07±0.24▲ | 0.91±0.07▲ |
| CIFAR-10 ResNet-110 | Accuracy | 94.48±0.10 | 94.27±0.39 | 94.44±0.17 | 93.21±0.28 |
| | ECE w/o post | 3.92±0.12 | 3.32±0.15 | 4.20±0.12 | 2.48±0.20 |
| | ECE with TS | 0.65±0.09 | 1.50±0.12▲ | 1.61±0.11▲ | 0.99±0.20▲ |
| | | Cross-Entropy | Label Smoothing 0.09 | $L_1$ Norm 0.01 | Focal Loss 5 |
| CIFAR-100 ResNet-32 | Accuracy | 67.23±0.20 | 67.50±0.30 | 65.91±0.41 | 64.99±0.62 |
| | ECE w/o post | 18.40±0.15 | 1.21±0.31 | 8.20±0.18 | 3.39±0.45 |
| | ECE with TS | 1.24±0.25 | 1.32±0.27▲ | 4.01±0.81▲ | 2.00±0.43▲ |
| CIFAR-100 ResNet-110 | Accuracy | 75.32±0.45 | 75.79±0.36 | 75.82±0.31 | 74.10±0.18 |
| | ECE w/o post | 13.45±0.16 | 3.82±0.33 | 14.56±1.87 | 8.06±0.42 |
| | ECE with TS | 1.61±0.11 | 3.30±0.20▲ | 3.45±0.81▲ | 1.81±0.23▲ |

Table I: Comparison results (mean±std) of several calibration metrics on CIFAR-10 over 5 random runs.

| | Cross-Entropy | Label Smoothing 0.05 | $L_1$ Norm 0.05 | Focal Loss 3 |
|---|---|---|---|---|
| NLL w/o post | 0.468±0.017 | 0.366±0.017 | 0.428±0.024 | 0.372±0.004 |
| NLL with TS | 0.294±0.009 | 0.366±0.017▲ | 0.428±0.023▲ | 0.370±0.004▲ |
| Brier w/o post | 0.158±0.004 | 0.154±0.006 | 0.161±0.010 | 0.181±0.002 |
| Brier with TS | 0.141±0.004 | 0.153±0.006▲ | 0.160±0.010▲ | 0.180±0.002▲ |
| SCE w/o post | 1.33±0.04 | 0.82±0.04 | 0.91±0.01 | 0.78±0.04 |
| SCE with TS | 0.47±0.02 | 0.79±0.04▲ | 0.86±0.03▲ | 0.64±0.03▲ |
| ACE w/o post | 0.68±0.03 | 0.57±0.03 | 0.61±0.01 | 0.54±0.03 |
| ACE with TS | 0.28±0.02 | 0.52±0.04▲ | 0.61±0.02▲ | 0.44±0.04▲ |

[2] Alex Krizhevsky. Learning multiple layers of features from tiny images. In *Tech Report*, 2009.

[3] Ken Lang. Newsweeder: Learning to filter netnews. In *International Conference on Machine Learning*, pages 331–339, 1995.

[4] Jishnu Mukhoti, Viveka Kulharia, Amartya Sanyal, Stuart Golodetz, Philip Torr, and Puneet Dokania. Calibrating deep neural networks using focal loss. In *Advances in Neural Information Processing Systems*, pages 15288–15299, 2020.

[5] Yuval Netzer, Tao Wang, Adam Coates, Alessandro Bissacco, Bo Wu, and Andrew Y. Ng. Reading digits in natural images with unsupervised feature learning. In *Advances in Neural Information Processing Systems Workshops*, 2011.

[6] Adam Paszke, Sam Gross, Francisco Massa, Adam Lerer, James Bradbury, Gregory Chanan, Trevor Killeen, Zeming Lin, Natalia Gimelshein, Luca Antiga, Alban Desmaison, Andreas Köpf, Edward Yang, Zachary DeVito, Martin Raison, Alykhan Tejani, Sasank Chilamkurthy, Benoit Steiner, Lu Fang, Junjie Bai, and Soumith Chintala. Pytorch: An imperative style, high-performance deep learning library. In *Advances in Neural Information Processing Systems*, pages 8024–8035, 2019.

[7] Jeffrey Pennington, Richard Socher, and Christopher D. Manning. Glove: Global vectors for word representation. In *Empirical Methods in Natural Language Processing*, pages 1532–1543, 2014.