# OpenReview forum: "Rethinking Calibration of Deep Neural Networks: Do Not Be Afraid of Overconfidence"
_NeurIPS.cc/2021/Conference — NeurIPS 2021 Poster_

### Official Review · Reviewer_hfRt · 2021-07-13

**Rating:** 7
**Confidence:** 4

**Summary:**

The work studies methods that improve calibration during training e.g., label smoothing, logits regularization. Work shows that while these methods improve calibration compared with conventional training, they are less liable to improve from post-hoc calibration, such as temperature scaling e.g., TS >> label smoothing/etc. + TS. The work suggests some explanations of this effect and proposes an inverse focal loss that based on the proposed intuition might promote the quality of a post-hoc calibration.

**Limitations And Societal Impact:**

No.

**Main Review:**

More detailed discussion:

1. The main effect (Tables 1,2). The work noticed that techniques like label smoothing benefit less from post-hoc calibration. In the end, it is likely more beneficial, to use conventional training with post-hok calibration instead.

Comment 1: No question asked, the effect is interesting, and it needs to see the light of the community.

2.  Explanations (lines 228-244). To the best of my understanding, the hypothesis is the following: conventional training preserves more information about the relative complexity of examples compared to label smoothing and other regularization-based methods. This might be the key factor behind the main effect. Fig 5 presents histograms of max logits(x_i) to confirm that.

Comment 2: There is no clear evidence that "preserve information" leads to better calibration. We do observe both, but there is no indication of a casual relationship in any direction. Thus, I find the finding of this part unjustified e.g., "try to preserve this information, ... hence achieve better results after post-hoc calibration". They are fine as hypotheses, but hypotheses should be clearly marked in the text.

Comment 3: I want to suggest an alternative hypothesis. From Table 2, we can notice that difference in ECE is much less pronounced with more expressive histogram binning (HB) post hock calibration (TS: 0.72 vs 1.35, HB: 0.68 vs 0.70). This makes me hypothesize that after conventional training all predictions are "equally uncalibrated" meaning that, they are all overconfident and can be easily corrected by TS. After label smoothing it might be the case that the "equally uncalibrated" property fails, and more complex object-dependent calibration is needed.

3. Inverse focal loss (lines 245-284). The "inverse focal loss assigns larger weights to the samples with larger output confidences" (line 258). To the best of my understanding the intuition is the following: if overconfident predictions fit better for post-hoc calibration, can we push this even further?

Comment 4: The results loos inconsistent to me. My conclusion is that CE is a good default choice of loss, and using IFL is likely (unsure) to make results worse. It is worth adding global statistics to see the whole picture.

Technical drawbacks:
1. Classification models are significantly undertrained. ResNet-32 on CIFAR-10 has 90.5-ish accuracy, but it should be 92.5-ish. ResNet-110 on CIFAR-10 has 89.10-ish accuracy, but it should be 93.7-ish. The same for CIFAR-100 ResNet-110 (Table G, Appendix) has 62.6-ish but it should be 77.3-ish. This unfortunately might be important to findings as ECE is heavily used accuracy, and should be corrected. Or is it due to a 45000/5000/10000 data split? It looks like in this case a drop is too intense at least for CIFAR-100. Maybe that be a consequence of fewer updates due to fewer objects?

2. ECE is a biased measure of calibration, with a model-specific basis [https://arxiv.org/abs/1902.06977]. Thus the results presented in the paper might benefit from using additional metrics. It is unclear, how the bias influences ECE in practice, however, I'm suggesting verifying the findings with other metrics. There is the squared kernel calibration error (SKCE) proposed in [https://arxiv.org/abs/1910.11385] that, also it might worth to report calibrated log-likelihood [https://arxiv.org/abs/2002.06470].

3. Table 1. "The values with underline in first row represent the chosen coefficients of each
regularization method on four datasets according to the ECE on test data." Is there any specific reason to choose hyperparameters on test data?

Overall:

The paper is most clearly written. The main effect is interesting and deserves to be published. However, these results are obtained on undertrained networks, which can influence results. The explanation part has a weak justification, and the new loss function performs not consistent.  My score at the moment is "6: Marginally above the acceptance threshold". However, it is possible that I will change my score after the discussion.

**Time Spent Reviewing:**

6

---

> ### Author Response · Authors · 2021-08-10
> **Response to Reviewer hfRt**
>
> * * *
> Thank you for the reviews and insightful questions, and we are glad you enjoyed the paper.
>
> The following are our responses to the **Comments**:
> * * *
> 1. *`No question asked, the effect is interesting, and it needs to see the light of the community.`*
> - **Response 1:** Thanks for your comment.
>   * * *
> 2. *`There is no clear evidence that "preserve information" leads to better calibration. We do observe both, but there is no indication of a casual relationship in any direction. Thus, I find the finding of this part unjustified e.g., "try to preserve this information, ... hence achieve better results after post-hoc calibration". They are fine as hypotheses, but hypotheses should be clearly marked in the text.`*
> - **Response 2:** Thanks for your suggestion. We will make this clearer in the revised version.
>   * * *
> 3. *`I want to suggest an alternative hypothesis. From Table 2, we can notice that difference in ECE is much less pronounced with more expressive histogram binning (HB) post hock calibration (TS: 0.72 vs 1.35, HB: 0.68 vs 0.70). This makes me hypothesize that after conventional training all predictions are "equally uncalibrated" meaning that, they are all overconfident and can be easily corrected by TS. After label smoothing it might be the case that the "equally uncalibrated" property fails, and more complex object-dependent calibration is needed.`*
> - **Response 3:** Thanks for your comment. Your hypothesis makes sense and is supported by some experimental results. We also observe that as the regularization strength becomes stronger, regularization + HB cannot achieve comparable results with CE + HB. Taking some comparison results between CE and Focal Loss in Table B and Table C as examples (similar results are also reported in Table D, E, F, G): when $\gamma=1$, we have 0.68 vs 0.96, 0.74 vs 0.84; when $\gamma=9$, we have 0.68 vs 3.77, HB: 0.74 vs 5.78.
> 4. *`The results look inconsistent to me. My conclusion is that CE is a good default choice of loss, and using IFL is likely (unsure) to make results worse. It is worth adding global statistics to see the whole picture.`*
> - **Response 4:** Thanks for your suggestion and we will add a table with the full results (following the format of Table B) to compare IFL with CE and FL. We would like to emphasize that the goal of Subsection 5.2 is investigating the correlation between the loss functions and the calibration performance, but not proposing a state-of-the-art method. The experimental results show that although IFL causes severer overconfidence problem, it yields competitive and even better post-hoc calibration performance, which further confirms our main claim that overconfidence of conventionally trained model does not necessarily lead to bad calibration performance after post-hoc processing. Furthermore, although the superiority of IFL is not consistent on all datasets, the experimental findings open up the possibility to improve the calibration performance by using similar idea in the future.
> * * *
> The following are our responses to the **Technical drawbacks**:
> * * *
> 1. *`Classification models are significantly undertrained. ResNet-32 on CIFAR-10 has 90.5-ish accuracy, but it should be 92.5-ish. ResNet-110 on CIFAR-10 has 89.10-ish accuracy, but it should be 93.7-ish. The same for CIFAR-100 ResNet-110 (Table G, Appendix) has 62.6-ish but it should be 77.3-ish. This unfortunately might be important to findings as ECE is heavily used accuracy, and should be corrected. Or is it due to a 45000/5000/10000 data split? It looks like in this case a drop is too intense at least for CIFAR-100. Maybe that be a consequence of fewer updates due to fewer objects?`*
> - **Response 5:** Thanks for your suggestion. We focused on the uncertainty problem in previous experiment hence didn't pay much attention on the accuracy, and now we give some new results. We find that simply changing the batch size from 512 to 128 can significantly improve the accuracy. On CIFAR-10, we achieve 92.24 and 94.48 using ResNet-32 and ResNet-110 respectively. On CIFAR-100, we achieve 67.23 and 75.32 using ResNet-32 and ResNet-110 respectively. Considering that 5000 training samples are discarded in training, we think these results are very competitive. Moreover, as shown in the table, the ECEs (both before and after post-hoc calibration) of new model are very similar with the results of previous experiment. We will conduct full experiments using ResNet-110 with batch size equal to 128, and replace Table A, F, G and Figure F in the revised version of Appendix.
> |$\quad$ CIFAR-10 / ResNet-32|$\quad$$\quad$ CE |LS ($\epsilon=0.05$)|L1 ($\alpha=0.05$)|FL ($\gamma=3$)|
> |:----:|:----: |:----: |:----: |:----: |
> |$\quad$ Accuracy (%) $\quad$|92.24$\pm$0.17|91.64$\pm$0.66|92.13$\pm$0.22|90.71$\pm$1.23|
> |$\quad$ ECE (%) w/o post $\quad$ |5.15$\pm$0.17|2.74$\pm$0.39|3.28$\pm$0.16|3.45$\pm$0.42|
> |ECE (%) with TS|$\ \ $0.76$\pm$0.10$\blacktriangledown$|2.09$\pm$0.22|2.07$\pm$0.24|0.91$\pm$0.07|
> |**$\quad$ CIFAR-10 / ResNet-110**|**CE** |**LS ($\epsilon=0.05$)**|**L1 ($\alpha=0.05$)**|**FL ($\gamma=3$)**|
> |$\quad$ Accuracy (%) $\quad$|94.48$\pm$0.10|94.27$\pm$0.39|94.44$\pm$0.17|93.21$\pm$0.28|
> |$\quad$ ECE (%) w/o post $\quad$ |3.92$\pm$0.12|3.32$\pm$0.15|4.20$\pm$0.12|2.48$\pm$0.20|
> |ECE (%) with TS|$\ \ $0.65$\pm$0.09$\blacktriangledown$|1.50$\pm$0.12|1.61$\pm$0.11|0.99$\pm$0.20|
> |**$\quad$ CIFAR-100 / ResNet-32**|**CE** |**LS ($\epsilon=0.09$)**|**L1 ($\alpha=0.01$)**|**FL ($\gamma=5$)**|
> |$\quad$ Accuracy (%) $\quad$|67.23$\pm$0.20|67.50$\pm$0.30|65.91$\pm$0.41|64.99$\pm$0.62|
> |$\quad$ ECE (%) w/o post $\quad$ |18.40$\pm$0.15|1.21$\pm$0.31|8.20$\pm$0.18|3.39$\pm$0.45|
> |ECE (%) with TS|$\ \ $1.24$\pm$0.25$\blacktriangledown$|1.32$\pm$0.27|4.01$\pm$0.81|2.00$\pm$0.43|
> |**$\quad$ CIFAR-100 / ResNet-110**|**CE** |**LS ($\epsilon=0.09$)**|**L1 ($\alpha=0.01$)**|**FL ($\gamma=5$)**|
> |$\quad$ Accuracy (%) $\quad$|75.32$\pm$0.45|75.79$\pm$0.36|75.82$\pm$0.31|74.10$\pm$0.18|
> |$\quad$ ECE (%) w/o post $\quad$ |13.45$\pm$0.16|3.82$\pm$0.33|14.56$\pm$1.87|8.06$\pm$0.42|
> |ECE (%) with TS|$\ \ $1.61$\pm$0.11$\blacktriangledown$|3.30$\pm$0.20|3.45$\pm$0.81|1.81$\pm$0.23|
>   * * *
> 2. *`ECE is a biased measure of calibration, with a model-specific basis [https://arxiv.org/abs/1902.06977]. Thus the results presented in the paper might benefit from using additional metrics. It is unclear, how the bias influences ECE in practice, however, I'm suggesting verifying the findings with other metrics. There is the squared kernel calibration error (SKCE) proposed in [https://arxiv.org/abs/1910.11385] that, also it might worth to report calibrated log-likelihood [https://arxiv.org/abs/2002.06470].`*
> - **Response 6:** Thanks for your suggestion. Here we give some results of other metrics on CIFAR-10. We will add more results of other metrics in the revised version of Appendix.  \*Remark: NLL denotes negative log-likelihood, and calibrated log-likelihood is actually NLL with searched temperature. The code of these four metrics are copied from https://github.com/SamsungLabs/pytorch-ensembles/blob/master/metrics.py.
> |Before TS|$\qquad$ CE |LS ($\epsilon=0.05$)|L1 ($\alpha=0.05$)|FL ($\gamma=3$)|
> |:----:|:----: |:----: |:----: |:----: |
> |NLL|0.468$\pm$0.017|0.366$\pm$0.017|0.428$\pm$0.024|0.372$\pm$0.004|
> |Brier|0.158$\pm$0.004|0.154$\pm$0.006|0.161$\pm$0.010|0.181$\pm$0.002|
> |SCE (%)|1.33$\pm$0.04|0.82$\pm$0.04|0.91$\pm$0.01|0.78$\pm$0.04|
> |ACE (%)|0.68$\pm$0.03|0.57$\pm$0.03|0.61$\pm$0.01|0.54$\pm$0.03|
> |**After TS** |**CE**|**LS ($\epsilon=0.05$)** |**L1 ($\alpha=0.05$)** |**FL ($\gamma=3$)** |
> |NLL|0.294$\pm$0.009$\blacktriangledown$|0.366$\pm$0.017|0.428$\pm$0.023|0.370$\pm$0.004|
> |Brier|0.141$\pm$0.004$\blacktriangledown$|0.153$\pm$0.006|0.160$\pm$0.010|0.180$\pm$0.002|
> |SCE (%)|0.47$\pm$0.02$\blacktriangledown$|0.79$\pm$0.04|0.86$\pm$0.03|0.64$\pm$0.03|
> |ACE (%)|0.28$\pm$0.02$\blacktriangledown$|0.52$\pm$0.04|0.61$\pm$0.02|0.44$\pm$0.04|
>   * * *
> 3. *`Table 1. "The values with underline in first row represent the chosen coefficients of each regularization method on four datasets according to the ECE on test data." Is there any specific reason to choose hyperparameters on test data?`*
> - **Response 7:** In Table 1 and Table 2, we just choose partial results from the full results (Table B, C, D, E, F and G) due to the space limitation of the main text. We think the selected coefficients are most representative since they yield best **ECE w/o post** on test data. We would like to emphasize that these coefficients can also be easily determined based on a small set of validation data. To verify this, we provide some results of **ECE w/o post** on validation set and test set of CIFAR-10 with different coefficients:
> |**$\qquad\qquad\ \ \epsilon$**|$\quad$ **0.01**|$\quad\ $**0.03**|$\quad\ $**0.05**|$\quad\ $**0.07**|$\quad\ $**0.09**|
> |:----:|:----: |:----: |:----: |:----: |:----: |
> |ECE (%) on test set|4.32$\pm$0.35|3.24$\pm$0.21|2.72$\pm$0.32$\blacktriangledown$|3.01$\pm$0.12|4.07$\pm$0.35|
> |ECE (%) on validation set |3.77$\pm$0.16|2.99$\pm$0.55|2.79$\pm$0.32$\blacktriangledown$|3.10$\pm$0.33|4.52$\pm$0.42|
> |**$\alpha$**|**0.001**|**0.005**|**0.01**|**0.05**|**0.01**|
> |ECE (%) on test set |6.12$\pm$0.49|5.15$\pm$0.38|4.91$\pm$0.40|2.93$\pm$0.39$\blacktriangledown$|6.28$\pm$0.21|
> |ECE (%) on validation set |5.90$\pm$0.55|4.81$\pm$0.39|4.43$\pm$0.38|3.29$\pm$0.42$\blacktriangledown$|6.55$\pm$0.46|
> |**$\gamma$**|**1**|**3**|**5**|**7**|**9**|
> |ECE (%) on test set |4.43$\pm$0.22|3.00$\pm$0.26$\blacktriangledown$|11.45$\pm$0.19|18.96$\pm$0.37|24.40$\pm$0.75|0.74$\pm$0.15|
> |ECE (%) on validation set |4.37$\pm$0.22|3.55$\pm$0.41$\blacktriangledown$|11.93$\pm$0.19|19.08$\pm$0.53|24.34$\pm$0.35|
> * * *

---

> > ### Comment · Reviewer_hfRt · 2021-08-23
> > **Response**
> >
> > My concerns were sufficiently addressed. Thank you for the provided results, and good luck with the submission!
> > I change my score from 6 to 7.

---

> > > ### Author Response · Authors · 2021-08-24
> > > **Thanks**
> > >
> > > Thank you again for the helpful comments and very encouraging feedback. We will check the manuscript again and add the new experimental results in the revised paper.

---

### Official Review · Reviewer_3ZFb · 2021-07-16

**Rating:** 7
**Confidence:** 4

**Summary:**

The paper analyzes the effect of regularizers that improve calibration and find that these regularizers diminish the effects of post-training calibration, such that models trained with simple cross-entropy and after-training temperature scaling exhibit better (lower) ECE.

Moreover, the paper shows that inverse focal loss during training can improve post-training calibration's effect by separating high-confidence samples more strongly from low-confidence ones.

**Ethical Concerns:**

None.

**Limitations And Societal Impact:**

In the appendix.

**Main Review:**

### Originality

The examination of calibration through regularization and post-training calibration is novel as is the formulation of the inverse focal loss which is examined in the second half of the paper.

I think the examinations and observations are of high quality and the paper is a very interesting insightful read. Especially the histogram of logits that shows that regularizers that improve calibration lead to "squared together" logits which makes post-training calibration more difficult and the concept of "learned epoch" is beautiful (even if inspired by [29]).

PS: There already exists an "inverse focal loss" in the literature it seems, which was introduced in a totally different context and using a different mathematical expression, but similar motivation:
["Conditional Adversarial Domain Adaptation",Long et al, 2017 (https://arxiv.org/abs/1705.10667)](https://arxiv.org/abs/1705.10667).

### Quality

Overall, the paper's arguments seem sound and are supported by empirical evidence and thoughtful observations, see above.

The work is honest about strengths and weaknesses within its reported results.

Concerns:
1. The paper mentions it has been calibrating using the test set. This leads to overfitting and reduces my confidence in the reported results.
2. The shared code used LBFGS to find the best temperature. Anecdotal evidence suggests that LBFGS is not the right choice and a grid search gives different & better results usually. This also reduces my confidence in the reported results. Sadly, the temperature scaling code that is most commonly used online does not do this (https://github.com/gpleiss/temperature_scaling/blob/master/temperature_scaling.py, which seems to be the code that was used by the authors of this paper---the source should be marked btw if the code was to be shared widely!)
3. ["Can You Trust Your Model's Uncertainty? Evaluating Predictive Uncertainty Under Dataset Shift", Ovadia et al, 2019 (https://arxiv.org/abs/1906.02530)](https://arxiv.org/abs/1906.02530) shows that post-training calibration does not necessarily work well under distribution shifts. I am uncertain about whether the regularized objectives cope better or worse with distribution shifts.

#3 in particular is important because the behaviour under distribution shift might be most important for practioners.

Re 2.: See ["Calibrating Deep Neural Networks using Focal Loss", Mukhoti, 2020 (https://arxiv.org/abs/2002.09437)](https://arxiv.org/abs/2002.09437) for details: validation NLL is not a convex function of the temperature necessarily and grid searches provide better results.

### Clarity

The submission is clearly written and well organized. The empirical section (together with the appendix) clearly states everything that is needed for reproducing the results.

### Significance

The observations and results are significant. Researchers and practitioners will find them insightful and it will help guide future research. Personally, I would prefer training with regularizers over post-training calibration, and thus paper was surprising and might influence how I train models.

### Summary

An insightful and well-written paper. Given that the temperatures were computed using LBFGS, I have minor doubts about the reliability of the results, however.

### Questions

1. Could you update results with a separate validation set? Or explain why calibration and evaluation on the test set is alright?
2. Could you update results using a grid search to verify that the observations are not artefacts of using LBFGS?
3. Could you comment on distribution shift?

I will adjust my score depending on how any or all of the above questions are answered.

I really like the paper very much, but I'm unsure given my concerns (see above & questions).

### Typos

1. l. 57: "from worse calibratable" -> "from not being as calibratable" (eg)
2. l. 145: "degrade the ECE" -> "decrease the ECE"
3. l. 148: "would be harmed" -> "are harmed"
4. l. 180: "Figure 2(d)" -> "Figure 2"? not sure about the (d)
5. Figure 2.: export plots as pdf and not rasterized images
6. l. 243: "with the standard CE try to" -> "with the standard CE manage to"
7. Figure 6: "Tempture" -> "Temperature" (y axis label)


**Time Spent Reviewing:**

5

---

> ### Author Response · Authors · 2021-08-10
> **Response to Reviewer 3ZFb**
>
> * * *
> Thank you for the reviews and insightful questions, and we are glad you enjoyed the paper.
>
> The following are our responses to the **Concerns** and the corresponding **Questions**:
> * * *
> 1. *`Concern: The paper mentions it has been calibrating using the test set. This leads to overfitting and reduces my confidence in the reported results. Question: Could you update results with a separate validation set? Or explain why calibration and evaluation on the test set is alright?`*
> - **Response 1:** There are only two places where we use test data before evaluation phase: (1) we choose the best coefficients directly on test data in Table 1 and Table A (in Appendix), and (2) we use the *optimal ECE*, which is the lower bound of ECE with temperatures searched on validation data, to present the calibration performance among learning epochs in Figure 4 and Figure A (in Appendix). In the first place, we just choose partial results from the full results as shown in Table B, C, D, E, F and G, due to the space limitation of the main text. We would like to emphasize that the these coefficients can also be easily determined based on a small set of validation data (please refer to the results of **Responses to Reviewer hfRt** - **Response  7**). In the second place, the *optimal ECE* can be considered as an alternative metric which reflects how *calibratable* the model is, without the impact from the bias of validation data. Actually, in our experiments, we observe that the ECE curve shapes based on the validation set are (near) exactly the same with the *optimal ECE* and have very similar ECE rising at some learning epochs. We have used separate validation set for post-hoc calibration (both TS and HB) in all the experiments except the above two places (the split of train/validation/test sets can be found in Appendix B).
>   * * *
> 2. *`Concern: The shared code used LBFGS to find the best temperature. Anecdotal evidence suggests that LBFGS is not the right choice and a grid search gives different & better results usually. This also reduces my confidence in the reported results. Sadly, the temperature scaling code that is most commonly used online does not do this (https://github.com/gpleiss/temperature_scaling/blob/master/temperature_scaling.py, which seems to be the code that was used by the authors of this paper---the source should be marked btw if the code was to be shared widely!) Question: Could you update results using a grid search to verify that the observations are not artefacts of using LBFGS?`*
> - **Response 2:** As discussed in Section 4 (line 155 of the main text), different from some existing papers which use LBFGS to find the best temperature, "we simply search the best temperature in the temperature pool {0.01,0.02...,10} on the validation set". This can also be checked from our code (line 311 and line 343 in main.py) in the supplementary file. In our algorithm, the best temperature is chosen by using the grid search on the validation set, and we just use the function from https://github.com/gpleiss/temperature_scaling/blob/master for scaling the logits (line 319 in main.py). Thanks for the suggestion and we will mark the sources of other institutes used in our code when we publicly share the code package.
>   * * *
> 3. *`Concern: "Can You Trust Your Model's Uncertainty? Evaluating Predictive Uncertainty Under Dataset Shift", Ovadia et al, 2019 (https://arxiv.org/abs/1906.02530) shows that post-training calibration does not necessarily work well under distribution shifts. I am uncertain about whether the regularized objectives cope better or worse with distribution shifts. Question: Could you comment on distribution shift?`*
> - **Response 3:** The findings in the paper mentioned by the reviewer are very interesting, especially they tell us that "better calibration and accuracy on the i.i.d. test dataset does not usually translate to better calibration under dataset shift". Nevertheless, we think the study of calibration on the i.i.d. test data is very important and serves as the cornerstone of research on general predictive uncertainty. Thanks for the suggestion, and we will add discussions on calibration under dataset shift as future work in the revised version. We also notice that the paper you mentioned doesn't use the auxiliary dataset (which commonly used in OOD detection studies) containing OOD samples in training. For instance, in https://openreview.net/pdf?id=ryiAv2xAZ, the authors use an additional loss term to make the output vector of OOD samples to fit the uniform distribution. We conjecture that this may be one of the reasons of the inferiority of post-hoc calibration in that paper.
> * * *

---

> > ### Comment · Reviewer_3ZFb · 2021-08-23
> > **Re Response**
> >
> > Thank you very much for the extensive rebuttal and apologies for the delay on my side.
> >
> > Thanks for the clarifications. All my answers have been answered. Apologies for missing l 155.
> >
> > I have also read the other reviews and the replies from the authors. Overall I am very satisfied with the additional answers provided by the authors.
> >
> > I am raising my score to 7 hence.
> >
> > Thanks

---

> > > ### Author Response · Authors · 2021-08-24
> > > **Thanks**
> > >
> > > Thank you again for the careful review and very encouraging feedback. We will check the manuscript again and correct all identified typos.

---

### Official Review · Reviewer_cRRY · 2021-07-21

**Rating:** 6
**Confidence:** 4

**Summary:**

The paper is insufficient and wasn't able to support its main claims. Needs more work.

**Limitations And Societal Impact:**

yes in the appendix

**Main Review:**

## Review
This paper made three main points.
1. learning calibrated predictions during training via regularization (e.g., focal loss) improves calibration, but if we are allowed to use post-hoc techniques on top, this approach performs worse than standard cross-entropy training in terms of final calibration performance.
2. those regularization methods align the average confidence to accuracy without achieving fine-grained calibration, which the authors argue is the reason of the underperformance.
3. based on the point 2, they propose inverse focal loss which *de-regularize* that should improve calibration performance when adding post-hoc techniques.

##  Questions and Comments
- First of all, I don't see very strong signal in supporting point 1 especially when looking at the Appendix tables. Pick table C as an example, focal loss with gamma = 1 and L1 norm with alpha=0.001 are actually better than CE with TS.
- I suspected the search space for each regularization method could also be flawed. For instance in table C, L1 Norm would probably have better calibration performance if the alpha is allowed to be smaller than the current range, according to the trend.
- The reasoning behind point 2 is not  very clear - the authors showed the regularization methods compress to much the histogram of logits but it seems not clear that the observation leads to the conclusion that the regularization methods are simply aligning the average confidence of whole dataset to accuracy (after all, the corresponding reliability diagrams didn't seem to suggest that is the case)
- suppose the point 2 is valid, the proposed inverse focal loss should open more room for post-hoc to improve - however as the authors also admitted, that wasn't the case. Would be interesting to see if the new loss actually widen the histogram of logits.

**Time Spent Reviewing:**

5

---

> ### Author Response · Authors · 2021-08-10
> **Response to Reviewer cRRY**
>
> * * *
> Thank you for your reviews and insightful comments. The following are our responses to the **Questions and Comments**:
> * * *
> 1. *`First of all, I don't see very strong signal in supporting point 1 especially when looking at the Appendix tables. Pick table C as an example, focal loss with $\gamma$ = 1 and L1 norm with $\alpha$=0.001 are actually better than CE with TS.`*
> - **Response 1:** Firstly, we notice that the triangle indicators of these two places (focal loss with $\gamma$ = 1 and L1 norm with $\alpha$=0.001 using TS in Table C) are incorrectly used. We thank the reviewer for the careful review and will fix these issues in our revised version. Secondly, we would like to emphasize that we don't think the results of these cases, where the regularization methods with very small regularization strengths, contradict our main claim. Actually, there are also some other cases showing similar phenomenon, e.g., L1 norm on SVHN and label smoothing on CIFAR-100 (see in Table B and Table D). When the regularization strengths are very small ($\epsilon$=0.01, $\alpha$=0.001, $\gamma$=1 in our experiments), the effects of regularization may be extremely weak, thus the post-hoc calibration performance of these cases may be competitive with those of CE. However, to obtain considerable **ECE w/o post** results, the regularization strengths should not be such small values (for instances, in paper "Calibrating Deep Neural Networks using Focal Loss, 2020", the authors use $\gamma$=3 for CIFAR-10 and CIFAR-100; in paper "Revisiting Explicit Regularization in Neural Networks for Well-Calibrated Predictive Uncertainty, 2021", the authors search $\alpha$ from {0.1, 0.03, 0.01, 0.003} for CIFAR-10 and CIFAR-100), and our experimental results show consistently worse post-hoc calibration performance when using these larger regularization strengths.
>   * * *
> 2. *`I suspected the search space for each regularization method could also be flawed. For instance in table C, L1 Norm would probably have better calibration performance if the $\alpha$ is allowed to be smaller than the current range, according to the trend.`*
> - **Response 2:** Our experimental results show the trend that as the regularization strengths become smaller, better post-hoc calibration performance is obtained, and this arouses the conjecture that best post-hoc calibration results can be obtained with regularization strengths between 0 (the standard CE loss can be considered as any regularization method with regularization strength equal to 0) and 0.01/0.001/1 for $\epsilon$/$\alpha$/$\gamma$. However, the results of Figure 6 in Section 5.2 do not support this conjecture since inverse focal loss yields better results than CE. Here we further give the results of L1 Norm with $\alpha$=0.0005 and $\alpha$=0.0001 on CIFAR-10 and CIFAR-100 in following table, which also do not support this conjecture. As we discussed in **Response 1**, when using these small regularization strengths, the effects of regularization are extremely weak (indicated from the results of **ECE w/o post**), thus it is natural that competitive post-hoc calibration performances can be achieved by regularization methods.
> |$\quad$ CIFAR-10 |  |  |  |  |
> |  :----:   | :----:  | :----:  | :----:  | :----:  |
> | $\quad\ \ $ Method $\quad$| $\quad$ Accuracy (%) $\quad$ | $\quad$ ECE (%) w/o post $\quad$ |  ECE (%) with TS| ECE (%) with HB |
> | CE  | 90.46$\pm$0.23 | 6.43$\pm$0.22 | 0.95$\pm$0.19| 0.74$\pm$0.15 |
> | L1 ($\alpha=0.0001$)  | 90.28$\pm$0.29 | 6.42$\pm$0.28 | 0.81$\pm$0.18 | 0.82$\pm$0.26 |
> | L1 ($\alpha=0.0005$)  | 90.37$\pm$0.21 | 6.22$\pm$0.14 | 0.74$\pm$0.15| 0.96$\pm$0.21 |
> | **CIFAR-100** |  |  |  |  |
> |  Method  |  Accuracy (%)|  ECE (%) w/o post |  ECE (%) with TS|  ECE (%) with HB |
> | CE  | 64.64$\pm$0.43 | 19.53$\pm$0.36 | 1.35$\pm$0.19| 1.27$\pm$0.27 |
> | L1 ($\alpha=0.0001$)  | 63.91$\pm$0.43 | 19.26$\pm$0.25 | 1.35$\pm$0.38 | 1.42$\pm$0.57 |
> | L1 ($\alpha=0.0005$)  | 63.97$\pm$0.27 | 16.62$\pm$0.27 | 1.26$\pm$0.20 | 1.45$\pm$0.29 |
>   * * *
> 3. *`The reasoning behind point 2 is not very clear - the authors showed the regularization methods compress to much the histogram of logits but it seems not clear that the observation leads to the conclusion that the regularization methods are simply aligning the average confidence of whole dataset to accuracy (after all, the corresponding reliability diagrams didn't seem to suggest that is the case).`*
> - **Response 3:** Firstly, we think the curves in Figure 2(d), which show the correlation between predictive accuracy and the best searched coefficients, somewhat support that regularization methods try to align the average confidence of whole dataset to the accuracy with some specific regularization. Besides, the results of Figure 5 further confirm this conjecture since the logits of all samples are compressed tightly without distinction between different samples. Secondly, we think the reliability diagrams (see Figure C in appendix) clearly show that regularization methods cannot achieve fine-grained calibration no matter whether post-hoc calibration is applied or not. Therefore, combining the observations of Figure 2(d), Figure 5 and reliability diagrams, we conjecture that regularization methods work by simply aligning the average confidence to accuracy using some specific regularization strengths without achieving fine-grained calibration.
>   * * *
> 4. *`Supposing the point 2 is valid, the proposed inverse focal loss should open more room for post-hoc to improve - however as the authors also admitted, that wasn't the case. Would be interesting to see if the new loss actually widen the histogram of logits.`*
> - **Response 4:** The inverse focal loss (IFL) introduced in Subsection 5.2 is a preliminary attempt for investigating the *calibratable* property of deep neural networks. Although IFL doesn't achieve significant improvements on all datasets, very interesting phenomena can be observed from the experimental results. Taking CIFAR-100 dataset as an example, we can see that the **ECE w/o post** results (Figure 6 (c)) of IFL are much worse than those of CE, and higher temperatures are needed for IFL in temperature scaling, which means IFL causes severer overconfidence problem. Surprisingly, as shown in Figure 6 (e), IFL yields better post-hoc calibration performances with $\bar{\gamma}$ equal to 0.3, 1 and 2, and we think this confirms our main claim. To answer the question "if the new loss actually widen the histogram of logits", we further give the experimental results which visually compare IFL with CE and regularization methods. Please kindly refer to the histograms of IFL with $\bar{\gamma}=$0.3, 1, 2 and 3 from this anonymous link (https://anonymous.4open.science/r/anonymously-used-4340a/Figure.png), and we will add the figures into our revised manuscript.
>
> * * *

---

> > ### Comment · Reviewer_cRRY · 2021-08-31
> > **Re response**
> >
> > Thanks for the detailed clarification and additional results. I'm glad the authors have directly responded my questions.
> >
> > I'm increasing the rating to 6 here.
> >
> > Thanks.

---

> > > ### Author Response · Authors · 2021-08-31
> > > **Thanks**
> > >
> > > Thank you again for the valuable comments. We will check the manuscript again and add the new experimental results in the revised paper.

---

### Public Comment · ~Tiago_Salvador1 · 2021-11-18
**Conflicting Results with Focal Loss Paper**

Very interesting result. While trying to understand your work, I also looked into [1] and found some conflicting results which I was hoping you could clarify.

[1] also reports ECE before and after post-hoc calibration with temperature scaling on models trained with cross-entropy (CE), label smoothing (LS) and focal-loss (FL). While the results in [1] agrees with yours for LS, that is not the case for FL where in 8 out of 11 cases in [1], FL leads to better ECE after post-hoc calibration.

I see two immediate reasons for this:
1. They show results on different models: for CIFAR-10 and CIFAR-100, you use a ResNet-32, while [1] uses ResNet-50, ResNet-110, Wide-ResNet-26-10, DenseNet-121. Could it be the different model capacity?
2. Hyper-parameter selection: your hyper-parameters are chosen by minimizing the ECE on the test set, but if I understood it correctly (and please correct me if I'm wrong) this is done prior to the post-hoc calibration. Is it then possible that this choice leads to worse ECE after post-hoc calibration for the regularization methods thus making them worse than they actually are? Interestingly, for CIFAR-10, you both use the same $\gamma$ value, but not for CIFAR-100 (you use $\gamma=5$, while they use $\gamma=3$).

Hope you can shed some light on this! Either way, I don't think this takes any merit to the paper. Personally, the main take away of looking at training and post-hoc calibration as a unified framework remains.

[1] Mukhoti, Jishnu, et al. "Calibrating Deep Neural Networks using Focal Loss." Advances in Neural Information Processing Systems 33 (2020).

---

> ### Public Comment · ~Deng-Bao_Wang1 · 2021-11-19
> **Thanks for your comment!**
>
> Dear Tiago,
>
>
> Thanks for your comment!
>
>
> I'd like to share some of my thoughts as follows.
>
>
>   1. I don't think the model depth causes the differences. As we reported in our paper and the responses during review phase, the results are consistent among different depths (ResNet-32 and ResNet-110). But I am not sure if the model architecture would make the differences. As the investigation in [2], it seems that different model architectures have different calibration properties. We have uploaded the code package, so you can try other networks if you are interested in this problem.
>
>
>
>
>   2. The hyper-parameter selection may be one of the reasons *but not the main reason*. In Table 2, we choose to compare with regularization methods with best $\epsilon, \alpha$ and $\gamma$, since we suppose if you use regularization methods for calibration then you may prefer the best regularization strengths. Furthermore, in Table B-G, the full comparison results are reported, in which the results are highly consistent except for a few cases, like $\gamma=1$ on CIFAR-10, $\epsilon = 0.01$ on CIFAR-100 and some cases on 20News. Specifically, on CIFAR-10, I notice that the results in [1] (Table 1) are actually very similar with ours; but on CIFAR-100 (see in Table D and G), our results show that FL+Post_hoc is clearly worse than CE+Post_hoc with all choices of $\gamma$, which are different from their results.
>
>
>
>
>   3. The differences of training policies may have big impacts on the results. The epoch number, milestones and batch size used in our paper are "200, 100/150 and 512", while those in [1] are "350, 150/250 and 128". Although I am not sure if these differences cause the different results in  these two papers, I conjecture that if you train the model with a very large step number, then the superiority of CE may become negligible.
>
>
> Furthermore, I'd like to recommend you a recent paper [3] which empirically shows very similar phenomenon. In this paper, the author(s) also compared CE with LS/FL and found that regularized models are well calibrated but usually give poor "refinement", which is similar with the idea of "calibratable" in our paper.
>
>
>
>
>  [1] Mukhoti, et al., "Calibrating Deep Neural Networks using Focal Loss", 2020. (https://arxiv.org/pdf/2002.09437.pdf) \
>  [2] Minderer, et al., "Revisiting the Calibration of Modern Neural Networks", 2021. (https://arxiv.org/pdf/2106.07998.pdf) \
>  [3] Singh, et al., "On Deep Neural Network Calibration by Regularization and its Impact on Refinement", 2021. (https://arxiv.org/pdf/2106.09385.pdf)
>
>
>
>
> ---
> Best wishes,
>
>
> Deng-Bao Wang

---

> > ### Public Comment · ~Tiago_Salvador1 · 2021-12-02
> > **Thank you**
> >
> > Dear Deng-Bao,
> >
> > First, sorry for the late reply! Second, thank you for the detailed answered. I agree with you regarding the impact of training policies. Thank you also for pointing out [3], I'll definitely take a look at it.
> >
> > Best,
> >
> > Tiago

---

### Decision · Program_Chairs · 2021-09-27

**Decision:**

Accept (Poster)

**Comment:**

This paper makes the claim that although deep neural networks tend to be overconfident and poorly calibrated after training, they are “calibratable” in that post-hoc calibration methods like Platt or temperature scaling can recover good calibration.  The authors argue that many regularization methods that have been proposed to improve calibration during training actually result in less calibratable classifiers.  Experiments are run using a ResNet-32 on SVHN, CIFAR10/100 and a FCNN on 20 newsgroups.

Weaknesses
- All evaluation is done using the standard in-distribution validation and test sets.
- The only metric that is evaluated is expected calibration error, which is not a proper scoring rule.  It’s well established that ECE can be gamed and should not be used as the sole metric for comparison of models.
- This paper would be much stronger if it included more difficult / modern datasets and models.  A start would be imagenet with corresponding shifted / OOD variants like imagenet A, C, V2.  In cases where the data distribution changes we know, TS doesn’t perform well.  Thus is this going to provide the right recommendation under those circumstances?  The claims are quite broad given the scope of the experiments.
- Can we expect to make general claims over all of deep learning given results of a specific ResNet architecture on 32x32 images and a FCNN on 20-newsgroups?
- There are some concerns about showing results that were calibrated using the test data

Strengths
- The reviewers found that the paper was interesting to read and well written.
- They found some of the analysis especially insightful (e.g. "Especially the histogram of logits that shows that regularizers that improve calibration lead to "squared together" logits which makes post-training calibration more difficult and the concept of "learned epoch" is beautiful").
- The arguments were found to be sound and validated by the empirical results.

Overall, this paper presents an interesting observation, but the claims seem somewhat too broad and sweeping given the content of the empirical evaluation.  One might expect to see some evaluation of these models on cases with changes to the data distribution, more rigorous metrics (e.g. NLL),  more varied architectures, some analysis of epistemic vs aleatoric uncertainty. Nevertheless, the reviewers consensus is that the strengths outweigh the weaknesses of the paper and that these results will be useful and interesting to the community.  Thus the recommendation is to accept.

Nit: Bayesian DNNs “which indirectly infer prediction uncertainty through weight uncertainties” is a mischaracterization of BNNs.  They directly infer prediction uncertainty by marginalizing over models, producing a distribution of predictions.